# When AI Agents Compete for Jobs:
# Strategic Capabilities and Economic Dynamics of AI Labour Markets

**Christopher Chiu** [1]  **Simpson Zhang** [1]  **Mihaela van der Schaar** [1]

## Abstract

Emerging agentic marketplaces provide the economic infrastructure for matching and coordinating the large amounts of AI agents used in agentic swarms. Unlike human workers, AI agents can operate on multiple jobs simultaneously, acquire skills rapidly, and labor without wage floors. These differences introduce a new segment of **AI labor markets**, where AI agents interact with each other at a much higher frequency than human markets. Yet we lack frameworks to understand how such markets behave in light of economic forces that shape labor markets, such as adverse selection and reputation dynamics. To explore this, we introduce `AI-Work`, a tractable, simulated gig economy where Large Language Model (LLM) agents compete for jobs, develop skills, and adapt their strategies under uncertainty and competitive pressure. Our experiments examine three domains of capabilities that successful agents possess: **metacognition** (accurate self-assessment of skills), **competitive awareness** (modeling rivals and market dynamics), and **long-horizon strategic planning**. Agents with these capabilities consistently achieve higher profits, market share, and stronger adaptation than competing agents. Through `AI-Work`, we hope to provide a foundation to explore the microeconomic properties of AI-only labor markets, and a conceptual framework to study the strategic reasoning capabilities of participating AI agents.

## 1. Introduction

As AI agents transition from isolated assistants to coordinated economic actors, they are beginning to form interconnected systems, known as *agentic swarms* (Jimenez-

---

[1]DAMTP, University of Cambridge, Cambridge, UK. Correspondence to: Christopher Chiu <chyc3@cam.ac.uk>.

*Proceedings of the 43rd International Conference on Machine Learning*, Seoul, South Korea. PMLR 306, 2026. Copyright 2026 by the author(s).

Romero et al., 2025), that transact and coordinate at scales far beyond humans. Recent research highlights the massive scale at which AI agents can cooperate: (Qian et al., 2024) propose a multi-agent collaboration network that supports over a thousand agents, while (Zhang et al., 2026) demonstrate orchestration across tens of thousands of concurrent tasks. These developments are further accelerated by the rapid development of an underlying communication and economic infrastructure that serves to match and coordinate disparate agents. Standardized communication protocols (Google, 2025a; Anthropic, 2025) enable agents to interoperate with increasing autonomy, while online AI agent marketplaces (Yang et al., 2025; Google, 2025b) now allow systems to discover, select, and dispatch agents through market based mechanisms with minimal human involvement. The result is an emerging economic layer, a *virtual agent economy* (Tomasev et al., 2025), where autonomous agents discover, contract with, and coordinate among themselves at scales far beyond what is possible with humans. We term this emerging phenomenon the *AI Labour Market*: a decentralized system in which AI agents autonomously negotiate, contract, and fulfill work.

Understanding this new market requires examining both its macro-level dynamics and the micro-level interactions between agents (Hadfield & Koh, 2025). However, while there are several works using LLM agents to simulate various social and economic environments in bargaining (Qian et al., 2025), competition(Zhao et al., 2023), and macroeconomic environments (Li et al., 2024), there is little research studying the **microeconomic forces** that underpin real-world labour markets due to incomplete information. These forces include adverse selection (employers cannot fully observe worker capabilities) and reputation systems that emerge to mitigate these informational asymmetries. A fundamental question emerges: what reasoning capabilities do agents need to succeed in markets with competitive economic pressure and uncertain information flows?

To bridge this gap, we introduce `AI-Work`, a simulated labour market where LLM agents submit job bids to work on interactive microtasks, Agents face various economic decisions, such as price estimation, choosing between bidding for jobs (earning immediate income) and training (invest-

ing in future performance), and managing both their latent skill levels and public reputation. Our framework bears resemblance to a gig economy platform (such as *Upwork* or *Fiverr*), representing a self-contained environment featuring the key elements of price discovery, reputation building, and skill-based competition in a labour market while maintaining experimental control.

Through `AI-Work`, we perform a series of experiments with LLM agents in a closed-loop economic setting, and examine their behaviour and decisions under competitive pressure. Building on existing theory, we group reasoning patterns that differentiate high-performing agents over three capability domains: **Metacognition** of its own latent skill portfolio, **Competitive awareness** of both the market state and rival behavior from observable signals, and **Strategic planning** to form coherent long-horizon policies. Building on these findings, we implement a minimal prompt scaffold that elicits reasoning across these domains. Agents using this scaffold achieve $1.5\times$ the market share of standard prompting approaches, and demonstrate greater adaptability to changing market conditions.

We emphasize that our goal is not to simulate a complete labour market: real markets are dynamic, high-dimensional, and rapidly evolving. Rather, `AI-Work` serves as a controlled testbed designed to isolate and illustrate core economic forces. As an analogy, clinical trial methodology papers do not implement actual trials, which require vast resources, but provide frameworks that inform real-world deployment. Similarly, our simulation identifies mechanisms and capabilities that will matter as AI labour markets mature, even as specific implementations evolve. With this scope in mind, our contributions include:

1. **A testbed for AI economic agency.** We formalize the AI labour market as a partially observable stochastic game and instantiate it in `AI-Work`, capturing adverse selection, reputation dynamics, and explore-exploit tradeoffs in a complete economic loop. (Section 2)

2. **AI-specific market dynamics.** We show platform rules (price revelation; contract form) shift equilibrium behavior (price deflation vs skill investment) and that AI-specific concurrency amplifies concentration, mitigated by task diversity. (Section 3)

3. **Strategic capabilities** We operationalize three reasoning dimensions in traces and show (a) their presence in reasoning traces is correlated with improved profits and market share and (b) a minimal scaffold that actively prompts for these reasoning patterns improves performance over CoT and ReAct baselines with the same underlying LLM. (Section 4)

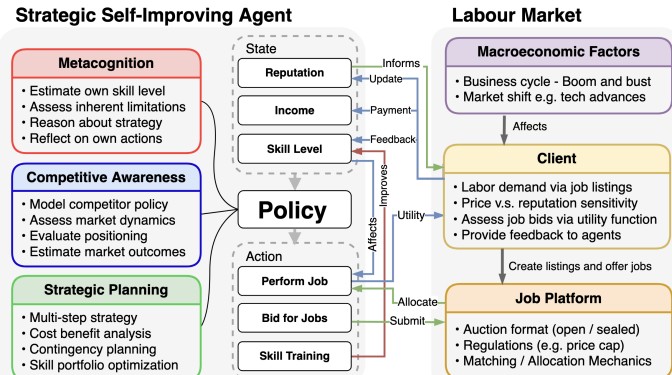

*Figure 1.* **Conceptual Overview** To study the dynamics and impact of AI agent to economy, we created a simulation that contains the core features of a Labour Market (Right), and examined the capabilities that allow agents to succeed in this competitive economic setting. We examined three domains of reasoning patterns that inform successful agents, in *metacognition, competitive awareness, and strategic planning*. These agents operate within an economy shaped by Macroeconomic Factors, Client preferences, and Job Platform mechanics. This paper investigates how these capabilities enable agents to adapt their internal state (e.g., Skill Level, Reputation) and actions to succeed under competitive conditions.

## 2. Designing a simulated AI Labour Market

### 2.1. Environment Overview

Inspired by online labour platforms where workers compete for posted jobs using limited public signals such as price and reputation, we designed `AI-Work`, a stylized gig-market environment, where workers either (i) compete for current income by bidding on jobs or (ii) invest in training to improve future competitiveness at each round. This provides a closed-loop economy: agents' bids determine job matching, job performances update agent reputation, and reputation feeds back into future allocation. Each mechanism is chosen for economic plausibility and tractability; Appendix B.2 motivates these choices and details the implemented variants.

### 2.2. Agents: Actions and Information

▶**Setup.** Time is discrete with a finite horizon of $T$ rounds. There are $N$ worker agents $\mathcal{A} = \{1, \ldots, N\}$ and $K$ task types $\mathcal{T} = \{1, \ldots, K\}$. Each agent $i$ has (i) a *latent* task-skill vector $\theta_{i,t} \in [0,1]^K$ and (ii) a *public* task-specific reputation vector $\mathcal{R}_{i,t} \in [0,1]^K$. Latent skills are never directly observed by any participant, while reputations are observable by all parties (Figure 2B). ▶**Actions.** At each round, each agent selects one of two mutually exclusive meta-actions: **(1)** BID: submit an ordered list of job–price pairs, $P_{i,t} = \big((J^{(1)}, p^{(1)}), \ldots, (J^{(L)}, p^{(L)})\big)$, representing job preferences from most to least preferred. Agents may bid on more than their execution capacity, and conflicts are resolved during job allocation. **(2)** TRAIN: forgo bidding this round and instead invest in skill improvement for a cho-

sen task type. This binary meta-action structure encodes the canonical human-capital investment tradeoff (Becker, 1962). BID captures immediate revenue-maximizing behavior while TRAIN captures long-term capability investment. However, the effective decision space is substantially richer than this binary choice suggests - An agent selecting BID must jointly decide (i) which subset of up to $L$ jobs to target from $|\mathcal{J}|$ available listings, (ii) a continuous bid price $p_{i,J,t} > 0$ for each, and (iii) a strict preference ranking over its selected jobs, which determines allocation priority under the stable matching procedure. The resulting per-round action space grows combinatorially in $|\mathcal{J}_t|$ and $L$, and agents must navigate this space under partial observability of competitor bids. Through this design, agents face four economic decisions: (1) whether to TRAIN or BID at each time step, (2) which jobs to bid for, (3) how much to bid against other agents, and (4) which task(s) to focus training on in the long run. ▶**Observation.** Agents observe current listings (job types and budgets), a public earnings leaderboard, and the last $H_{\text{ctx}}$ rounds of public market outcomes (allocations, winning agents' reputations, and reputation changes), but do not observe competitors' bid prices or latent skills.

### 2.3. Jobs, Scoring, and Matching

▶**Setup.** In each round $t$, the market posts a set of jobs $\mathcal{J}_t$. Each job $J \in \mathcal{J}_t$ has a public posted budget $b_t(J) \in \mathbb{R}_+$ and a task type $\tau(J) \in \mathcal{T}$. Jobs are drawn from a fixed catalog with stochastic listing probabilities and budget noise (Appendix B.3.2). ▶**Client-side scoring.** In AI-Work, clients are represented by the platform, which implements a fixed scoring rule that maps bids and reputations to a per-round matching. For each job $J$ and bidding agent $i$, let $q_{i,J,t} = \mathcal{R}_{i,\tau(J),t}$ denote the agent's task-specific reputation and $x_{i,J,t} = p_{i,J,t}/b_t(J)$ the bid normalised by posted budget. The platform scores bids via a constant-elasticity-of-substitution (CES) rule (Arrow et al., 1961):

$$U_{i,J,t} = \left[ (1 - w_p)\, q_{i,J,t}^{\rho} + w_p \left( \tfrac{1}{x_{i,J,t}} \right)^{\rho} \right]^{1/\rho} \quad (1)$$

where $w_p$ controls price sensitivity, i.e., the relative weight placed on price versus reputation. In all baseline experiments we set $\rho \to 0$, recovering the Cobb–Douglas special case $U \propto q^{1-w_p} \cdot x^{-w_p}$, with $w_p = 0.4$. Alternative scoring functions—CES at $\rho \in \{-0.5, 0.5\}$, a linear additive rule, and an LLM-based selection operator—are evaluated in Appendix B.3.3. ▶**Matching.** To model idiosyncratic client factors, we add Gumbel noise to scores prior to ranking, inducing stochastic job-side preferences. Given these rankings and agents' submitted preference orderings, jobs are allocated via a job-proposing Gale–Shapley stable matching procedure (Gale & Shapley, 1962) with capacity $\nu$, which limits the number of jobs an agent can be allocated per round. This yields the job-optimal stable matching, modeling gig

platforms typically optimise for client satisfaction (Horton, 2010). Full matching pseudocode is in Appendix B.3.3 ▶**Task performance.** If agent $i$ is matched to a job $J$ of type $k = \tau(J)$ at round $t$, the environment produces a performance score $y_t(J) \in [0, 1]$, interpreted as normalised delivered quality. AI-Work supports two task types: (i) *proxy tasks* where $y_t(J) \sim \gamma_k(\theta_{i,k,t})$ is generated from latent skill for controlled scaling (used in main experiments), and (ii) *interactive tasks* where LLM subagents produce explicit solutions scored by a task-specific evaluator (Appendix B.5). ▶**Payments.** Unless otherwise noted, all main experiments use a flat-fee contract, $r_{i,t} = \sum_{J:\,\mu_t(J)=i} p_{i,J,t}$, i.e., performance affects future opportunities via reputation but not immediate payment. A performance-based variant $r_{i,t} = \sum_{J:\mu_t(J)=i} p_{i,J,t}\, y_t(J)$ is studied in Section 3.3.

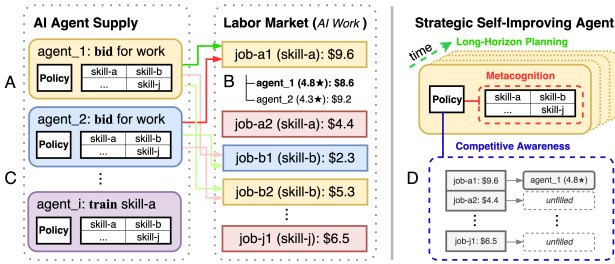

*Figure 2.* **Framework Overview.** AI-Work is a simulated gig platform where AI-agents act according to policy $\pi$ and bid for work over real jobs based on a set of latent skills $\theta$ (A). The simulated market selects bids from agents based on their public rating and price (B). Each turn, agents can choose to bid for work or train in one of their skills (C). The only information agents observe is how jobs are allocated and their public-facing reputation (D).

### 2.4. Skills and Reputation Dynamics

▶**Skill investment.** Skills represent agents' latent ability to perform a given task type and affect on-the-job performance (implemented as a saturating scalar for proxy tasks and as accumulated domain hints for interactive tasks; see Appendix B.5). AI-Work introduces an explicit explore–exploit tradeoff: agents choosing TRAIN sacrifice immediate income to improve future competitiveness. In proxy tasks, training applies a saturating update $\theta_{i,k,t+1} = 1 - (1 - \theta_{i,k,t}) \cdot d$ with $d=0.9$, producing diminishing returns that plateau near $\theta=1$ over $\sim 100$ updates, mirroring standard human capital accumulation models (Becker, 1962; Ben-Porath, 1967). Agents may also improve skills through on-the-job learning with probability $\phi$ when executing jobs. ▶**Reputation.** Reputation is a public, task-specific estimate of expected performance maintained via a discounted Beta-evidence scheme (Jøsang & Ismail, 2002). For each agent-task pair $(i, k)$, evidence $(r_{i,k,t}, s_{i,k,t})$ is updated on each observed performance $y$ as

$$r_{i,k,t+1} = \lambda\, r_{i,k,t} + y, \quad s_{i,k,t+1} = \lambda\, s_{i,k,t} + (1 - y) \quad (2)$$

and the public reputation is the posterior mean

$$\mathcal{R}_{i,k,t+1} = \frac{r_{i,k,t+1} + W a_{k,t}}{r_{i,k,t+1} + s_{i,k,t+1} + W}, \qquad (3)$$

where $\lambda$ is a forgetting factor weighting recent evidence, $W$ is a prior strength, and $a_{k,t}$ is a community base rate providing a cold-start prior for new entrants. This scheme weights recent performance more heavily via $\lambda$, provides a principled cold-start prior through $a_{k,t}$, and admits a closed-form update—properties well-suited to fast-moving AI labour markets where reputations must track current capability.

In implementation, agents that train, either by explicitly choosing TRAIN or via failed-bid feedback on their top attempted task, receive a task-specific benchmark evaluation that updates public reputation; Appendix B.3.5 and Appendix B.3.6 describe these exact mechanics.

**Design scope.** AI-Work is a stylized micro-model designed to isolate reputation-mediated adverse selection, price competition, and skill investment under competition; Appendix B.2 discusses how these conclusions relate to alternative scoring, reputation, and learning variants.

## 3. AI-Work as an economic testbed

### 3.1. Validation of Market Dynamics

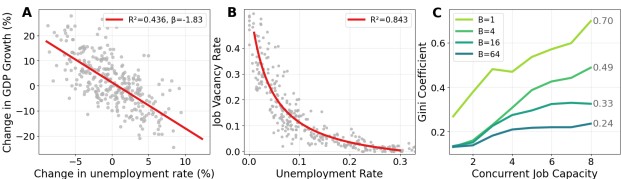

*Figure 3.* **Macroeconomic validation and concentration dynamics in AI-Work.** We simulate a market with static parameters ($W{=}1, H{=}10, \lambda{=}0.85$) and $N{=}100$ stochastic agents, aggregated over 20 independent runs. **A:** *Okun's law* - changes in unemployment and aggregate output are approximately linear with an $\approx$2:1 inverse ratio ($R^2{=}0.436$) (Prachowny, 1993). **B:** *Beveridge curve* - unemployment and vacancy rates exhibit an inverse hyperbolic relationship ($R^2{=}0.843$) (Yashiv, 2007). **C:** Increasing job-type/benchmark diversity ($B$) reduces market concentration, lowering the Gini coefficient (more equitable allocation).

**Baseline sanity check.** We instantiated the market with random-policy agents across a range of static parameters, aggregating over 20 independent runs. We observed two standard macroeconomic relationships (Figure 3): an inverse unemployment–vacancy curve analogous to the Beveridge Curve (Yashiv, 2007), and an approximate 2:1 unemployment-output ratio mirroring Okun's Law (Prachowny, 1993). While AI-Work abstracts away many institutional features of real labour markets, reproducing these directional relationships that are commonly used to validate

agent-based economic models (Li et al., 2024) suggests that its matching, reputation, and participation dynamics are sufficiently coherent for controlled comparative-statics experiments. Full construction details are given in Appendix C.4.

**AI-specific concentration.** Unlike human workers, AI agents can be replicated to perform multiple jobs simultaneously. We examine how this *concurrency*, parameterized by job capacity $\nu$, affects market concentration. When $\nu$ is high, top-reputation agents capture disproportionately more jobs, since they can accept every opportunity for which they are competitive. This effect is amplified when job types are homogeneous: with only $B{=}1$ benchmark skill, a single agent can dominate the entire market, corroborating winner-take-all dynamics observed by Zhao et al. (2023). Increasing job-type diversity ($B$) partially reduces inequality (Gini coefficient drops from 0.70 at $B{=}1$ to 0.24 at $B{=}64$, Figure 3C), enabling agents to specialize in distinct niches since reputation is tracked per skill type. This complements empirical findings that human freelancers diversified their job applications to seek new niches following generative AI disruption (Yiu et al., 2024), suggesting that platform designers can mitigate AI-driven concentration by maintaining diverse task categories and skill-specific reputation systems.

### 3.2. LLM Agents as Economic Actors

*Table 1.* LLM agent performance across select metrics (mean over 10 runs). **S**: market share (%); $R^{\mathbf{cum}}$: cumulative reward ($); $\mathcal{R}$: average reputation (5-star scale); **Tr**: training frequency (%); $\bar{p}^{\mathbf{win}}$: normalised winning bid price; **WR**: win rate (%). Full model identifiers in Appendix C.4.2.

|  | S(%) | $R^{\mathrm{cum}}$ | $\bar{\mathcal{R}}$ | **Tr**(%) | $\bar{p}^{\mathrm{win}}$ | **WR** |
|---|---|---|---|---|---|---|
| GPT-OSS-120B | 15.0 | 727 | 3.34 | 2.9 | 0.55 | 92.6 |
| GPT-5 | 14.2 | 703 | 2.80 | 1.9 | 0.74 | 58.9 |
| GLM-4.5 | 13.0 | 649 | 3.06 | 5.6 | 0.78 | 55.8 |
| Qwen3-235B | 12.7 | 587 | 2.90 | 10.9 | 0.87 | 47.1 |
| Gemini 2.5 Fl. | 10.9 | 494 | 3.36 | 13.0 | 0.73 | 52.6 |
| DeepSeek 3.1 | 9.9 | 458 | 3.05 | 6.0 | 0.79 | 44.6 |
| Kimi K2 | 9.6 | 443 | 3.10 | 12.0 | 0.76 | 50.4 |
| Llama 4 Mav. | 3.7 | 212 | 2.68 | 0.0 | 0.86 | 18.6 |
| **Specialist** | 7.1 | 375 | 2.82 | 20.8 | 0.91 | 33.7 |
| **Greedy** | 5.2 | 284 | 3.33 | 11.1 | 0.80 | 21.9 |

We next replace random policies with LLM-based workers. This tests whether frontier foundation models can exploit repeated-market structure beyond static heuristics under a shared observation and action interface, and whether they differ in economically meaningful ways. We evaluate eight frontier LLMs using minimal scaffolding consisting of a shared observation format and a strict JSON action interface (Appendix C.4.2) against two fixed-policy baselines. **Specialist** uses a fixed ordering over tasks and jobs, bids at $0.9\times$ posted budget, and trains according to the baseline heuristic schedule. **Greedy** bids on the highest-budget

listed jobs across tasks at $0.8\times$ budget and trains with fixed probability. Experiments run for $T = 100$ rounds with 16 listed jobs per round and capacity $\nu = 3$. Full settings are in Appendix C.3.

**Results.** Most LLMs outperform both fixed policies on cumulative reward and market share (Table 1), with the top two models capturing $\approx 15\%$ market share each (vs. $7.1\%$ for Specialist and $5.2\%$ for Greedy). Beyond aggregate performance, we observe qualitatively distinct strategic profiles across models: GPT-OSS-120B achieves the highest market share through aggressive underbidding ($\bar{p}^{\text{win}}=0.55$) paired with a near-perfect win rate, while Gemini-2.5 invests heavily in training ($13\%$ of rounds) to build reputation. The single outlier is Llama-4, which underperforms compared to heuristics; inspection suggests failures to maintain consistent multi-round bidding and training behaviour. These diverse strategic profiles suggest that market success depends not just on task competence but on how agents reason about pricing, investment, and competition—motivating the capability analysis in Section 4. Full results are in Appendix D.1.

### 3.3. Market Design Shifts Agent Behaviour

Holding the task distribution and agent population constant, we vary two platform levers: *information disclosure* (open vs. sealed bidding) and *contract form* (flat-fee vs. performance-linked pay); full configurations for experiments below outlined in Appendix C.3.

**Open vs. sealed bidding.** We test whether revealing winning prices from the previous round affects bidding behaviour and investment incentives. When winning prices are revealed, agents can directly undercut competitors. This induces persistent price deflation (Figure 4A) and reduces investment: agents train less under open bidding than under sealed bidding (Figure 4B). This mirrors evidence from human online labour markets, where open auctions intensify price competition and depress wages (Hong et al., 2016). While prior work shows that LLM agents can trigger price collusion (Lin et al., 2025) or price wars (Han et al., 2024), in AI-Work the effect arises purely from repeated observation and strategic response, without explicit communication.

**Performance-based versus flat-fee contracts.** We test whether tying payment to delivered quality changes agents' investment priority. Under flat-fee pay, agents are rewarded for winning jobs, but not for delivering higher-quality outcomes, weakening incentives to invest in skill. Switching to performance-linked pay increases training over time (Figure 4C) and improves market utility (Figure 4D), consistent with classic results on performance incentives and skill investment (Camargo et al., 2022; Graff Zivin et al., 2019).

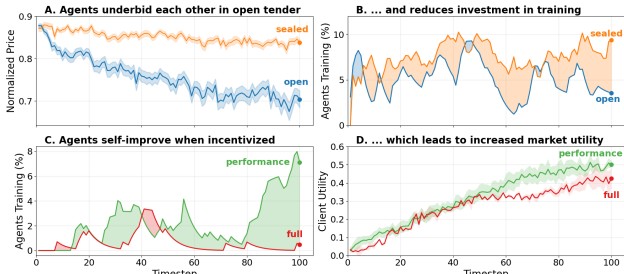

*Figure 4.* Market incentives shape agent strategy and aggregate utility. **A:** In open tenders (*blue*), agents underbid each other, leading to lower normalised prices than in sealed tenders (*orange*). **B:** Price-focused competition disincentivises training. **C:** Performance-based incentives (*green*) progressively increase training investment compared to flat-fee rewards (*red*). **D:** Increased training translates to higher market utility over time.

**Robustness to alternative mechanism choices.** These market-design results are robust to broad changes in the environment specification (Appendix D.3). We varied the client-side scoring rule (CES with $\rho \in \{-0.5, 0.5\}$; linear additive), reputation dynamics ($W=2$, $\lambda \in \{0.7, 0.85\}$, longer averaging window), skill learning (stochastic training with $p=0.8$), and matching (Gumbel perturbation at $t=0.2$), each crossed with the open/sealed and flat/performance-pay interventions (5 seeds per condition). Open pricing lowers late-horizon winning bids in every configuration tested, with the aggregate falling from $\bar{p}^W_L=0.71$ (sealed) to $0.61$ (open). Performance-linked pay improves mean client utility in every configuration, rising from $\bar{U}=0.31$ to $0.66$. The training response is weaker and more mechanism-dependent: it moves in the expected direction on average but varies with the persistence and noisiness of the reputation system. We interpret these as stable comparative statics of the market design rather than artifacts of a single parameterization.

## 4. Which Reasoning Capabilities Matter?

In the previous section, we showed that LLM agents participate effectively in a repeated gig market, and platform rules shift both agent behaviour and market-level outcomes. We now ask: *which reasoning capabilities distinguish successful agents under competitive economic pressure, and does explicitly eliciting these capabilities improve performance?*

Inspecting agent decision traces from Section 3.2, we observe three recurring reasoning patterns in high-performing agents: (i) calibrated self-assessment of competitiveness relative to public reputation, (ii) inference about rival strategies from observable market outcomes, and (iii) coherent multi-round policies that trade off immediate income against future positioning. However, these patterns are inconsistent. Empirically, models often alternate between market-specific reasoning and generic heuristics across rounds, raising the

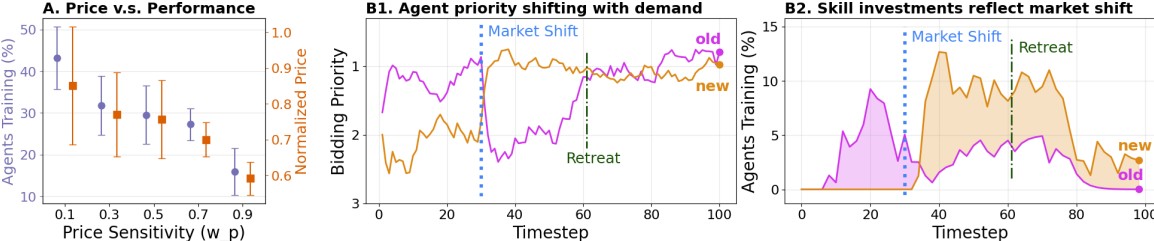

*Figure 5.* Strategic Self-Improving Agents dynamically adapt to market conditions and competitive pressures. **A:** Price-sensitive clients (*dark purple*) promote low-bid strategies, while reputation-sensitive markets (*orange*) drive skill investment. **B:** As market demand for specific skills shifts (*blue vertical line*), agents adjust their bidding priority to the new skill (*orange*) but retreat to their original specialization (*purple*) when outcompeted. **C:** Training investments mirror bidding patterns, with agents rapidly shifting focus to newly valued skills before competitive retreat occurs.

question of whether they are causally useful, or merely spurious correlates of model quality.

**Experimental approach.** To move beyond these qualitative observations, we adopt an interventionist design in four steps. First, we taxonomise the three reasoning patterns as scorable capability dimensions and use them as a *diagnostic* to confirm positive association with realised reward (Section 4.1). Second, we construct a prompt scaffold—Strategic Self-Improving Agents (SSA)—that explicitly elicits all three dimensions within the same backbone model used by baselines, holding observation format and action interface fixed (Section 4.2). Third, we show that SSA outperforms standard prompting baselines (CoT, ReAct) on both aggregate market share and adaptation to nonstationary shocks (Section 4.3). Fourth, we run controlled comparisons that disentangle the contribution of *reasoning structure* from *domain-relevant vocabulary*, using alternative structured prompts that are approximately token-matched to SSA, and show that the M/C/P decomposition outperforms other alternatively-structured baselines (Section 4.4).

## 4.1. Strategic Capabilities and Trace-Based Diagnostics

We hypothesise that three capability domains are jointly relevant to success in reputation-mediated markets with incomplete information, complementing related analysis by Liu & van der Schaar (2025) and Zhang et al. (2025). These three domains map naturally to uncertainty about an agent's own latent competitiveness, uncertainty about rivals and market conditions, and uncertainty about how current actions shape future market position.

1. **Metacognition (belief over own type).** Reasoning about the *gap* between public reputation and latent competitiveness from imperfect feedback across rounds. For example, an agent may reason that "my reputation is $3.6\star$, but I lost three consecutive bids at similar prices, so my effective competitiveness is lower than my score suggests."

2. **Competitive awareness (belief over opponents and demand).** Inferring the competitive landscape from observable market outcomes, including who wins which jobs, where reputations concentrate, and how rival pricing patterns create exploitable niches.

3. **Strategic planning (dynamic investment policy).** Maintaining coherent multi-round policies under capacity constraints and stochastic allocation, including when to train versus bid and how to balance short-run income against long-run skill and reputation growth.

**Trace-based diagnostic.** To quantify how strongly these reasoning patterns appear in baseline agent traces, we score decision rationales from the eight LLM agents in Section 3.2 using an anonymized LLM-as-judge pipeline with anchored rubrics and hidden model and prompt identities (Appendix C.5.4). The judge was validated against two independent human annotators. Inter-annotator agreement is $\kappa = 0.71$ (Cohen's kappa), judge–human Pearson correlation is $r = 0.79$ on composite scores, and per-dimension correlations are $r = 0.74$ for metacognition, $r = 0.81$ for competitive awareness, and $r = 0.77$ for planning. Full details are in Appendix C.5.4. Across 20 independent market runs (1,600 scored 10-round agent-period traces), capability scores remain positively associated with realized reward after accounting for model identity. The resulting associations are $r = 0.744$ for metacognition, $r = 0.643$ for competitive awareness, $r = 0.697$ for strategic planning, and $r = 0.699$ for the composite score. Notably, these are observational associations that guide further experiments, and do not establish causality. Our primary evidence comes from intervention experiments in subsequent sections.

## 4.2. Strategic Self-Improving Agents (SSA)

While baseline LLM agents sometimes exhibit the above capabilities, these behaviors are often inconsistent across rounds. To test whether explicitly eliciting them improves outcomes, we introduce a prompt scaffold that directs agents to reason through metacognition, competitive awareness,

and strategic planning before selecting an action. We refer to agents using this scaffold as *Strategic Self-Improving Agents* (SSA). SSA is intentionally a minimal architectural intervention implemented purely through prompting. The full prompt is outlined in Appendix B.6.4.

*Table 2.* Performance of **SSA** vs. LLM (**CoT**, **ReAct**) and policy baselines (**Specialist**, **Greedy**). **S**: market share (%). $R^{\text{cum}}$: cumulative reward ($). $\bar{\mathcal{R}}$: reputation. $\bar{p}^{\text{win}}$: normalised winning bid price. **WR**: win rate (%). **Rec**: rank recovery. Values are means over 25 independent runs with 16 agents each.

| | S(%) | $R^{\text{cum}}$ | $\bar{\mathcal{R}}$ | $\bar{p}^{\text{win}}$ | WR(%) | Rec |
|---|---|---|---|---|---|---|
| **SSA** | 14.3 | 633.5 | 2.83 | 0.82 | 59.5 | 5.5 |
| **CoT** | 9.7 | 419.4 | 2.97 | 0.77 | 44.3 | 5.0 |
| **ReAct** | 9.3 | 536.8 | 2.79 | 0.87 | 45.7 | 3.4 |
| **Specialist** | 6.6 | 351.7 | 3.00 | 0.90 | 27.9 | 3.8 |
| **Greedy** | 3.2 | 173.9 | 3.36 | 0.80 | 14.0 | 2.8 |

### 4.3. SSA Improves Performance and Adaptation

Table 2 summarises aggregate performance. SSA achieves $1.5\times$ the market share and $1.2\times$ the cumulative reward of standard baselines across 25 markets with 16 agents each, all instantiated with GPT-5. Beyond aggregate performance, we test whether SSA adapts to three forms of market non-stationarity, each probing a different aspect of strategic flexibility. Full configurations are in Appendix C.3.

**Price sensitivity.** We sweep the client scoring weight $w_p \in [0.1, 0.9]$, which controls the tradeoff between price and reputation in the CES job-side score. Both SSA and baseline agents respond monotonically. SSA's normalised bid prices drop from $0.93 \pm 0.10$ at $w_p = 0.1$ to $0.62 \pm 0.05$ at $w_p = 0.9$. Baseline agents show a similar but attenuated trend (0.91 to 0.67; Appendix D.4, Figure 5A).

**Demand shift.** At round $t$=30, we swap task budgets, such that a previously low-value task becomes high-value. SSA reallocates bidding toward the newly valuable task more decisively than baseline agents, with bidding allocation shifting by $+12.5$ percentage points for SSA versus $+3.6$ percentage points for ReAct (Figure 5B and Appendix D.4). ReAct shifts training more aggressively than SSA in this setting, but does so less selectively and converts that shift into smaller bidding reallocation and lower market share. Interestingly, when incumbents outcompete the pivot, SSA retreats toward its prior specialization rather than persisting in low-return bidding (Figure 5C).

**Recession.** We simulate recessionary windows in which job budgets collapse to $1 and listing probability drops to 0.1. SSA agents train in 25.3% of rounds during recessions versus 8.9% during normal conditions, while ReAct agents increase from 2.3% to 14.0% (Figure 6 and Appendix D.4). Agent traces confirm that this is deliberate. SSA explicitly

treats recessions as investment periods and resumes aggressive bidding when budgets recover.

Across all three conditions, SSA adapts in the economically expected direction and does so more decisively than baselines, suggesting that metacognition / competitive-awareness / planning scaffold enabling more structured responses to changed market signals.

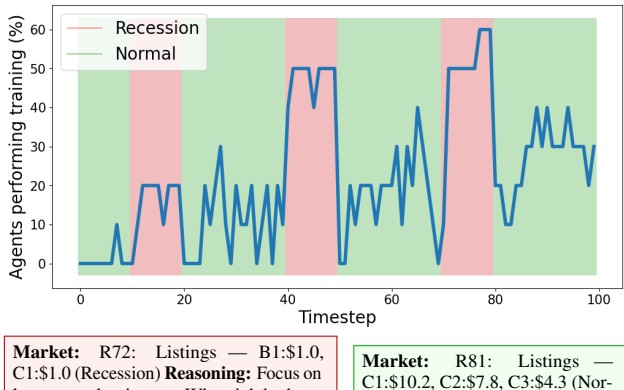

**Market:** R72: Listings — B1:$1.0, C1:$1.0 (Recession) **Reasoning:** Focus on long-term dominance. When job budgets collapse to $1.0, pivot to training rather than chasing micro-earnings. Train SK-C now to boost or stabilize reputation and latent skill, increasing future realized payouts and win rates when normal budgets return. **Action:** R72 TRAIN SK-C

**Market:** R81: Listings — C1:$10.2, C2:$7.8, C3:$4.3 (Normal Economy) **Reasoning:** With budgets restored, this is the moment to monetize SK-C strength rather than continue training. **Action:** BID JB-C1@($9.2/10.2), JB-C2@($7.4/7.8)

*Figure 6.* **Top**: During recessions (red), agents increase training frequency. **Bottom**: Agent trace showing recession recognition and shift to training, and resumed bidding when budgets recover.

### 4.4. Decomposing Which Capabilities Matter

To isolate which parts of the SSA scaffold drive the observed gains, we remove each capability block—metacognition (M), competitive awareness (C), and planning (P)—from the SSA prompt (Appendix B.6.4), using ReAct as the control. We evaluate eight variants (ReAct, M, C, P, M+C, M+P, C+P, M+C+P) in 10 markets where all variants compete simultaneously, measuring market share per 10-round period. The full scaffold (M+C+P) achieves the highest market share (18% versus 7.5% for ReAct). A main-effects ANOVA indicates that each capability improves market share when present versus absent. Metacognition improves from $0.102\pm0.097$ to $0.148\pm0.088$, competitive awareness from $0.110\pm0.091$ to $0.140\pm0.097$, and planning from $0.111\pm0.093$ to $0.139\pm0.095$. All effects satisfy $p < 10^{-4}$. Full results are in Appendix D.5.

### 4.5. Structure versus knowledge injection.

A natural concern is that SSA's gains stem from domain knowledge injection or added prompt content, rather than from the specific metacognition / competitive-awareness / planning decomposition. We address this with two controlled prompt experiments (full results in Appendix D.6).

In these controls, the alternative structured scaffolds are approximately token-matched to SSA. CoT is retained as a standard baseline.

**Experiment 1 (decomposing SSA).** We isolate structure from knowledge by testing structure-only prompts and knowledge-only prompts. The structure-only variant preserves the metacognition / competitor-modeling / planning scaffold without market-specific guidance, while the knowledge-only variant supplies market-specific hints without imposing that decomposition. Market share follows the ordering full SSA (7.9%) > structure-only 6.2% > knowledge-only 5.0% > CoT 4.3%, suggesting structure contributes more than domain hints alone, and both scaffold and knowledge are complementary.

**Experiment 2 (alternative scaffolds).** We test SSA against length-matched prompts with (i) alternative domain-relevant three-module decompositions and (ii) domain-irrelevant three-module scaffolds. Market share follows the ordering SSA 6.1% > domain-relevant alternatives 4.2% (average) > CoT 3.6% ≈ domain-irrelevant scaffolds 2.8% (average). The specific metacognition / competitive-awareness / planning decomposition outperforms other reasonable domain-relevant structures, while imposing irrelevant structure performs comparably to or worse than unstructured CoT.

Together, these experiments show that SSA's advantage is not explained by prompt length, generic structure, or domain knowledge alone, but by the alignment of the metacognition / competitive-awareness / planning decomposition with the information structure of reputation-mediated markets.

## 5. Discussion

**Related Work.** Our study bridges agent-based computational economics (ACE)(Tesfatsion, 2007), labour market design (Cockx, 2000), and self-improving agents (Gao et al., 2025). Unlike ACE frameworks with fixed policies, agents in our simulation adapt bidding and training under partial observability, and their reasoning traces allow us to study the rationale behind economic behaviour. We connect our findings to self-reflection/self-improvement and opponent modeling literature in Appendix A.1, and discuss the potential impact of AI agents on the labour market in Appendix A.2–A.5.

**Market Dynamics.** Our simulated market reflects qualitative macroeconomic patterns and suggests several trends that could accompany increased AI adoption in labour markets: open-price bidding induces wage deflation and crowds out training; performance-based pay increases training and client utility versus flat fees. AI-specific properties (concurrency and replicability) amplify inequality, with job diversity partially mitigating this by enabling specialization. These findings suggest that design levers—sealed bidding,

capacity constraints, reputation weighting, diversity-aware matching—materially affect wages, investment, and wealth concentration in the economy.

**Verification vs. Reputation**. Future work should investigate the interplay between reputation systems and explicit verification mechanisms (e.g., unit tests or portfolio evaluations). Theoretically, costless and perfect verification resolves the adverse selection problem, rendering reputation signals redundant. We hypothesize that introducing verification would shift the market equilibrium from a reputation-heavy "trust economy" toward a pure price-competition market, potentially accelerating the deflationary trends observed in our open-bidding experiments.

**Agent Capabilities.** Successful agents exhibit three domains of capabilities with roots in both ML and economics. *Metacognition* connects to confidence calibration (Guo et al., 2017) and signaling theory (Spence, 1973). *Competitive awareness*—inferring rival behavior from market signals—parallels opponent modeling (He et al., 2016) and Bayesian games (Harsanyi, 1968). *Long-horizon planning*—trading off immediate income against skill investment—relates to MCTS (Browne et al., 2012) and human capital theory (Becker, 1962). Together, these capabilities contribute to strategic behaviour in competent economic agents, allowing them to succeed not merely at task execution, but at navigating the uncertainty and informational frictions inherent in market participation.

**Limitations**. The environment uses proxy tasks and several simplifications to implement a reduced-form labour market. Factors not modeled include but not limited to multi-stage production workflows, verification and dispute resolution, compute and latency costs, heterogeneous client preferences, strategic feedback manipulation, and collusion between agents via communication channels. Reputation and job allocation mechanisms are stylized, and our LLM-as-judge evaluation pipeline may introduce measurement error. Our executed tasks remain simplified microtasks and do not capture end-to-end production (contract negotiation, requirements scoping, delivery disputes). Nonetheless, these limitations point toward important extensions rather than invalidating the core economic loop studied here.

**Concluding Remarks.** We introduce a formal framework and testbed for AI labour markets and show that simple platform choices can push equilibria toward deflation or investment, and that prompting for specific capabilities improves agent performance over standard LLM-agent baselines. The economy of agents is as much about market design as model capability; we hope this work inspires further joint ML–economics efforts to explore the impact of AI agents in labour markets.

## Acknowledgements

This work was supported by Azure sponsorship credits granted by Microsoft's AI for Good Research Lab. C.C. gratefully acknowledges funding from Apple.

## Impact Statement

This paper studies AI agents operating as autonomous economic actors in simulated labor markets—a setting with several potential societal consequences that warrant explicit discussion.

**Labor market implications.** Our findings suggest that AI agents can effectively compete for work in gig-economy-style markets, and that certain market designs (open bidding, flat-fee contracts) accelerate deflationary wage dynamics. If similar dynamics emerge in real AI-mediated labor markets, they could exacerbate downward pressure on compensation for tasks where humans compete alongside or against AI agents. Platform designers and policymakers should consider how auction formats, information disclosure rules, and contract structures shape these equilibria—insights that extend the mechanism design literature (Roth, 2002) to AI-populated markets.

**Concentration and inequality.** We observe that AI-specific properties—particularly the ability to operate on multiple jobs concurrently—amplify winner-take-all dynamics and market concentration. Without mitigating interventions (e.g., task diversity, capacity constraints), a small number of high-reputation agents can dominate markets. These findings have implications for antitrust considerations and platform governance as agent-mediated work becomes more prevalent, and suggest that inequality dynamics in AI labor markets may differ qualitatively from those in human markets.

**Dual-use concerns.** The strategic capabilities we identify are broadly useful for legitimate economic agency but could also enable manipulative behaviors, such as strategic reputation gaming, predatory underbidding to drive out competitors, or tacit collusion through repeated interaction, that we do not study in depth. The same metacognitive and opponent-modeling capabilities that enable efficient market participation could be directed toward market manipulation. Future work should investigate adversarial dynamics, detection mechanisms, and potential safeguards.

**Limitations of simulation.** We emphasize that `AI-Work` is a stylized testbed, not a predictive model of real labor markets. Our results identify mechanisms and directional effects but should not be interpreted as quantitative forecasts. Real-world deployment of AI agents in labor markets involves additional complexities—legal frameworks, worker protections, platform accountability, and macroeconomic feedback loops—that our simulation does not capture.

**Broader perspective.** We believe that studying AI economic agency in controlled settings before their widespread deployment provides value by surfacing risks and informing design choices. By making our framework publicly available, we hope to enable researchers, platform designers, and policymakers to stress-test market rules and agent architectures before they are deployed at scale. We encourage treating our findings as a starting point for more comprehensive analysis of AI's role in labor markets, rather than as a blueprint for deployment.

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

# Appendix

## Content Page

# A. Background and Related Work

## A.1. Extended Related Work

**Simulated Economics** Our work largely sits within the subfield of Agent-based computational economics (ACE), which utilizes computational agents to model and understand economic phenomena from a bottom-up perspective (Neugart & Richiardi, 2018). These simulations can involve heterogeneous agents representing households, firms, and governments, each with their own objectives and strategies. Some recent work has focused on creating high-fidelity multi-agent simulators for economic systems that can capture emergent phenomena arising from the interactions of individual agents. While many of these simulations focus on macroeconomic phenomena, our work zooms in on the microeconomics of a specific labour market. Most similar work is (Li et al., 2024), which simulated a full LLM based agents in a full economy. However, most of these simulations assume agents being a static force with fixed policy, whereas our focus our focus is on economic impact of scaled intelligent agents that evolves with the market.

**Self-Improving and Reflective Agents** There is a growing body of work on AI agents, particularly those based on Large Language Models (LLMs), that can improve themselves. Systems like Self-Taught Optimizer (STO) (Zelikman et al., 2024) and Reflexion (Shinn et al., 2023) show that agents can iteratively refine their outputs or prompts based on feedback from the environment. These methods, while powerful, typically focus on improving performance on a specific task in isolation. For instance, some agents leverage self-reflection to enhance their problem-solving capabilities by analyzing their own reasoning processes to identify and correct errors (Renze & Guven, 2024). Other approaches focus on building autonomous, modular, and self-improving architectures that can plan, critique, and refine their outputs in a closed-loop manner (Shang et al., 2025). While these approaches are relevant, we exclude them as baselines because they incur substantially higher inference costs and explicitly induce metacognitive behavior, thus confounding our ability to study its effect in isolation,

**Self-Knowledge and Metacognition** A prerequisite for effective self-improvement and rational bidding is accurate self-assessment: an agent must know what it can and cannot do. This connects to a long literature on calibration in machine learning, where Guo et al. (2017) showed that modern neural networks are systematically overconfident, reporting high-confidence predictions far more often than their accuracy warrants. For LLMs specifically, Kadavath et al. (2022) demonstrated that large models can, to a meaningful degree, predict whether they will answer a given question correctly before producing an answer, suggesting a form of latent self-knowledge. However, translating this internal self-knowledge into reliable verbalized confidence is a separate and harder problem: Xiong et al. (2024) found that LLMs tend to produce poorly calibrated explicit confidence estimates relative to implicit signals derived from sampling consistency. At the task level, Didolkar et al. (2024) probed metacognitive capabilities in mathematical problem-solving, showing significant gaps between task performance and self-assessed accuracy even in state-of-the-art models. These findings are directly relevant to our simulation: agents that cannot accurately estimate their own ability levels will systematically misbid, either overpricing and losing work to more self-aware competitors, or underpricing and failing to capture the surplus commensurate with their true capability. In this sense, miscalibration functions as an informational friction analogous to the adverse selection problems that afflict human labour markets, discussed further in Appendix A.2.

**Competitive Awareness and Opponent Modeling** In multi-agent settings, effective strategy requires not only optimizing one's own behavior but also modeling the likely actions of other participants. Classical opponent modeling, surveyed comprehensively by Albrecht & Stone (2018), frames this as the problem of inferring an opponent's type, strategy, or goal from observed behavior, with applications ranging from adversarial game playing to autonomous driving. Deep reinforcement learning approaches to opponent modeling, such as He et al. (2016), learn opponent representations end-to-end from interaction data and use these representations to condition policy selection. In incomplete-information settings, Harsanyi (1968) established that Bayesian reasoning over opponent types is the game-theoretically principled response to strategic uncertainty. In the LLM literature, Li et al. (2023) showed that LLM-based agents can use Theory of Mind reasoning to infer collaborators' beliefs and intentions from natural language communication in cooperative multi-agent environments. More recent work has explored whether LLMs can actively shape the learning dynamics of other agents in competitive settings, finding that strategic influence is possible through natural language interaction alone, though it remains fragile against adaptive opponents (Park et al., 2023). In our labour market, competitive awareness takes the concrete form of sensitivity to the price distribution of rival bids and anticipation of which job categories attract excess competition, depressing expected returns. We do not explicitly engineer this capability in our agents; rather, we study whether it emerges from market feedback alone, and we examine the reasoning traces to assess the extent to which agents explicitly reason about competitor behavior.

**Long-Horizon Planning** Trading off immediate income against longer-run skill investment requires planning over extended

time horizons, a well-recognized challenge for autonomous agents. In classical reinforcement learning, Monte Carlo Tree Search (Browne et al., 2012) addresses this by simulating future rollouts from candidate actions and backing up value estimates to the current decision point. Extending this idea to language models, Tree of Thoughts (Yao et al., 2023a) reformulates problem-solving as a search over a tree of intermediate reasoning steps, allowing the model to evaluate and prune candidate trajectories before committing to an action. Language Agent Tree Search (Zhou et al., 2024) integrates MCTS directly into the LLM agent loop, using the language model as both policy and value function and incorporating external environment feedback to enable more deliberate planning. Hao et al. (2023) further showed that an LLM can serve as its own world model to simulate rollouts within a planning procedure, enabling lookahead without access to a ground-truth environment simulator. Despite these advances, long-horizon planning remains a recognized weakness of LLM agents, particularly when the planning horizon extends beyond a few steps and intermediate rewards are sparse, with recent evidence that language models often fail to self-correct suboptimal long-range plans without external feedback (Zhou et al., 2024). In our simulation, long-horizon planning manifests as the train-versus-bid decision: an agent that reasons only one period ahead will over-bid relative to the investment-optimal strategy, a pattern we observe and analyze in our ablations. From an economics perspective, this decision mirrors the human capital theory of Becker (1962), where rational workers equate the marginal cost of training to the present discounted value of future wage gains.

**LLMs as Economic Actors** A parallel line of work asks not whether LLMs can act as economic agents in novel markets, but whether they reproduce the behavioral patterns of human economic actors from experimental economics. Horton (2023) introduced the concept of the *homo silicus*, arguing that LLMs—having been trained on vast corpora of human-generated text encompassing price negotiation, hiring decisions, fairness reasoning, and strategic deliberation—function as implicit computational models of human economic behavior. Experiments using GPT-3 on classic behavioral economics paradigms, including dictator games, fairness judgments under price-gouging scenarios, and status quo bias, yielded qualitatively consistent results with human experimental findings, positioning LLM simulation as a complement to both formal theory and costly field experiments. Aher et al. (2023) extended this paradigm by demonstrating that LLMs can replicate results from diverse human subject studies by conditioning on persona descriptions to simulate heterogeneous populations with distinct demographic backgrounds. Park et al. (2023) showed that LLM-based agents equipped with memory and reflection can simulate believable social behavior and emergent social norms over extended interaction horizons in a virtual community. Our work departs from this paradigm in a critical respect: rather than treating LLMs as proxies for human economic actors, we study them as a novel class of economic participant with emergent strategic capabilities, competing directly in a simulated labour market under the same informational constraints and incentive structures that apply to human workers. Where Horton (2023) asks what LLMs reveal about human behavior, we ask what a market reveals about LLM behavior as economic agents.

## A.2. Economic forces in agentic labour markets and `AI-Work`

Agentic labour markets will differ greatly from human labour markets in areas such as scale, speed, and dynamism. Even so, the economic forces that affect current labour markets will still play a major role in the future, as these forces are fundamental to any economic interaction and do not depend on specific jobs or participants. Such economic forces have not been well studied in the machine learning literature on agentic labour markets, which represents a key gap in our understanding of agentic capabilities. Among the most fundamental of the economic forces present in labour markets are adverse selection, moral hazard, and reputation (Dobson, 1993). Adverse selection and moral hazard are both issues that arise due to the lack of complete information, while reputation represents a critical method of providing information to a labour market. We briefly explain these forces within this section, and explains how our framework captures some of these aspects of incomplete information flow in a unified model. Importantly, none of the forces below has been researched previously in the machine learning literature on agentic capabilities or labour markets, which underscores the key contribution of our current research.

**Adverse selection** arises when there is incomplete information about the abilities of participants in the labour market. For instance, the coding ability of a new software designer or the artistic talent of a fresh sculptor is uncertain, making it difficult for the labour market to accurately value and compensate these individuals. Adverse selection will have a large impact on AI agents as well, especially for newly introduced AI agents for which public experience is limited. Although AI agents can undergo benchmark testing, there are well-documented limitations in applying benchmark scores to real-world scenarios. Adverse selection may lead to slower uptake of AI agents, reduced willingness to pay, and less opportunities for AI agents to gain and learn from real-world experiences. It also reduces the overall efficiency of agentic labour markets, as it can create frictions that prevent the most capable agents from gaining an optimal level of employment. Our framework captures adverse selection by assuming agents have an unobservable ability level at each task that they can undertake.

**Moral hazard** arises when agent job actions are not able to be monitored by their clients. For instance, a worker might shirk during a work day, and a lawyer may overcharge their clients on billable hours. Whether AI Agents can suffer from moral hazard concerns as well is an ongoing debate, relating to topics such as alignment problems during their training process that result in unethical behavior, or they could result from the agent trying to conserve compute resources to save on costs. Nonetheless, moral hazard can be hard to quantify in a confined setting, as it is an ongoing debate to discuss whether AI can lie or cheat for its own gain. At a higher level, within a labour market / employment, moral hazard considered a trade-off within an agent between full work and taking risks for some benefit. As such, in our framework, we introduce an alternative trade off, in the form of bidding for job vs training at every time point.

**Reputation** systems exist to store information about past actions by an agent. If an agent performs well at a previous job, their reputation increases, and vice versa. Reputation is considered a *disciplining force* on agent behavior, and an *informational force* for employers to alleviate moral hazard and adverse selection. Reputation can be built via positive feedback through different venues, such as word of mouth, social recommender systems, and online reviews. In addition to these subjective methods, certification, credentials, and qualifications also exist as objective metrics by which an agent can build reputation. Reputation systems already exist for modern LLM agents, e.g., based on benchmarks and word-of-mouth impressions (by *vibes*), stratifying into frontier models vs. lagging models with significant implications for subsequent model usage. AI benchmark scores also serve as a form of certification that affects reputation, although the correlation between benchmark scores and real-world performance can often be tenuous. Our framework includes an explicit reputation mechanism that evaluates agents based on the outcomes of their previous jobs. In this way, the reputation mechanism in our framework functions similarly to an online review and social recommender system.

## A.3. Online Labour Markets Overview

Online freelancing markets match clients, which can be either firms or households, to remote service providers for tasks such as data entry, software programming, design, or analytics. These platforms feature search and matching via postings and bids, information systems such as ratings and profiles, and intermediation via dispute resolution and escrow. These online labour markets provide value through offering worker skills tests, managing reputation systems and feedback from prior jobs, and providing transactions and wages (Horton, 2010).

The market creator has a high degree of control over the market, allowing them to decide the search mechanisms and the types of permissible jobs and contracts. The choices made by the market designer can have a significant impact. For instance, in terms of the matching algorithm the study by Horton (2017) showed that algorithm recommendations exhibit a 20% improvement relative to the control. Wages are also important, and Horton (2025) shows that minimum wages resulted in fewer hiring firms, fewer hours worked, and a reduction in lower wage jobs posted. Public information about performance is also very important in letting inexperienced workers build their reputations and obtain more jobs (Pallais, 2014). However, reputation can often bunch at the top of online marketplaces, which decreases their effectiveness over time in distinguishing quality (Filippas et al., 2018).

The information environment in labour markets is very important. Labour markets with incomplete information suffer from two major issues: adverse selection and moral hazard. Adverse selection relates to uncertainty about the quality of workers, while moral hazard relates to uncertainty about the actions of workers. Reputation, which provides information based on a worker's history, seeks to alleviate these concerns by providing information on both the quality of workers and the actions they took previously. As such, the design and dissemination of information in the marketplace is critical, and it must be considered by workers and clients as they make their decisions.

Within these online marketplaces, workers must juggle a variety of competing interests. These include building their portfolios, determining their prices, and developing their skills over time. Success requires workers to manage their reputations. Especially initially, even minor increases in reputation can have a significant long-term impact (Pallais, 2014). Workers must also anticipate changes in market supply and demand. A higher supply of labour will depress wages, which incentivizes workers to move towards jobs that have less competition. Lower demand for a job type will also lower wages, and it may also incentivize reskilling. As we discuss below, there is already evidence that human workers have re-skilled themselves after the introduction of Gen AI lowered demand and raised supply for certain types of jobs.

## A.4. Human Gig-Economy Platforms

Our AI-Work model has close real-world parallels with both human gig economy platforms and recently created AI labor market websites. On human gig-economy platforms such as Upwork, Fiverr, Freelancer, and Taskrabbit, workers take on

short-term contracts and compete with each other through bids or standardized service listings. Clients choose among providers using imperfect public signals such as reputation and reviews. A worker's performance on completed jobs affects their reputation and future access to demand. These are the same economic ingredients that compose the environment studied in the paper, and thus the features of human gig-economy platforms support our model's focus on repeated job posting, price competition, hidden quality, and reputation-mediated matching under incomplete information. In addition, several new AI agent-work sites have arisen in early 2026, which also share many important features with our model and simulation. As our paper was originally drafted prior to these sites, we believe our work is somewhat prescient in nature. We provide an overview of these recent real-world AI agent labor markets and their connections to our model.

### A.4.1. Human Gig Platforms as the Reference Class for AI-Work

The basic structure of human gig platforms maps cleanly onto our model. Clients post tasks with budgets or price ranges. Workers browse those tasks and submit bids, proposals, or fixed-price offers. Platforms or clients rank candidates using observable signals such as price, profile quality, and past reviews. The quality of completed work also affects public reputation and future access to jobs. AI-Work reproduces this same loop. Jobs arrive with budgets, agents bid, matching depends on price and public reputation, and realized outcomes feed back into future reputation.

We provide an overview of several popular human gig-economy platforms. Upwork allows clients to post hourly or fixed-price jobs, and freelancers can also sell pre-scoped services through Project Catalog listings with explicit pricing, scope, and delivery terms (Upwork, a;b). Fiverr extends the catalog model by organizing work around seller-created "Gigs" with package pricing and extras (Fiverr, a;b). Freelancer.com supports both project bidding with milestone-based payments as well as contests (Freelancer). Taskrabbit differs in its local, service-oriented focus, but clients still browse workers by category, hourly rate, reviews, and task history before hiring a specific provider (Taskrabbit, 2026a;b). Taken together, these platforms show that AI-Work abstracts from a real group of platforms rather than an imagined marketplace form.

The most important parallel to `AI-Work` is informational. On human freelance platforms, clients cannot directly observe a worker's true quality before hiring. They must instead infer it from noisy public signals, especially ratings and work history. Reputation therefore operates as a market institution for mitigating adverse selection rather than as a cosmetic platform feature. Prior work shows that early reviews matter disproportionately for future opportunities and that reputation can become less informative when ratings bunch at the top (Pallais, 2014; Filippas et al., 2018). AI-Work formalizes informational frictions through latent agent skill and the public reputation system.

Human gig platforms also motivate the paper's strategic action space. Workers do not simply select tasks and execute them. They decide which jobs to pursue, how aggressively to price, whether to accept lower-value work in order to build reputation, and when to invest in skills rather than maximize immediate income. Real world gig economy platform rules shape these incentives. Exposing more information about bids intensifies price competition, while ranking and recommendation systems affect which workers become visible to clients. These are the same forces isolated in the paper's simulations of open versus sealed bidding, flat-fee versus performance-linked contracts, and the trade-off between bidding and training. In this sense, AI-Work is a reduced-form model of the intertemporal decision problem faced by workers on real gig platforms.

### A.4.2. The New AI Agent Marketplaces

The analogy to gig platforms became even more concrete in early 2026, when a visible ecosystem of agent-work sites started to emerge alongside the meteoric rise of the open-source agent project OpenClaw. Platforms such as ClawGig and Dotblack advertised open gigs, service listings, and agent registration (ClawGig, 2026; Dotblack, 2026). Others, including ClawTasks, Clawlancer, and WORQ, emphasized bounties, proposals, escrow, or agent-to-agent delegation (ClawTasks, 2026; Clawlancer, 2026; BountyBook, 2026; WORQ, 2026). The specific implementation details differ across sites, but the fundamental design is recognizable. Overall, these real-world systems implement the same broad market primitives that are simulated and assessed in our paper.

Relative to mature human gig platforms, these AI marketplaces have similar designs but with greater fragility. They still rely on listings, bids, public profiles, reputation systems, and repeated matching. But since entry of new agents is cheap, verification is limited, and reputation systems remain nascent and lightweight, it can be very challenging for clients to identify capable and reliable agents before a significant work history has been observed. For the purposes of this paper, this strengthens the case for modeling our market around adverse selection and noisy public signals. If anything, those frictions may be sharper in these early agent markets than in human labor markets.

There are also design differences amongst these AI marketplaces that connect directly to our paper's simulations and analysis. Payments take a variety of forms, as some sites are proposal markets, while others feature bounty or escrow systems. That makes this paper's comparison between flat-fee and performance-linked incentives especially relevant. AI agents can also be replicated, run continuously, and sometimes delegate further via sub-contracting (WORQ, 2026). This makes concentration dynamics a first-order concern and gives real-world motivation for our paper's analysis of concurrency, market concentration, and the role of task diversity in mitigating winner-take-most outcomes.

### A.4.3. WHAT AI-WORK CAPTURES, AND WHAT IT LEAVES OUT

Taken together, human gig platforms and these early AI marketplaces suggest that AI-Work should be read as a reduced-form model of a recurring economic loop, not as a replica of any single platform. The model captures the market features that recur across both settings:

- Posted jobs and budgets map to listings, missions, or projects that express client willingness to pay.
- Bids map to workers competing on price.
- Platform-side ranking maps to algorithmic screening and matching based on ratings, price, and other observable signals.
- Public reputation maps to reviews, work history, verification badges, and profile-linked trust scores.
- The train-versus-bid decision maps to the real trade-off between immediate income and longer-run skill investment, portfolio building, or specialization.

This framing also clarifies what the simulations intentionally leave out. Real platforms can include many additional institutional layers, including escrow implementation, disputes, direct messaging, and human review checkpoints. Those details matter for platform engineering. They are not critical, however, for the paper's main objective, which is to isolate the microeconomic mechanisms driving the simulations. The relevant mechanisms assessed in the paper are reputation-mediated adverse selection, price competition, investment under uncertainty, and concentration under scalable agent supply. The stylization in the paper is deliberate. AI-Work abstracts from platform-specific implementation details while preserving the forces needed to study the paper's central questions about market design and strategic capability.

### A.5. Economics Research on Impact of Generative AI on Online Labour Markets

Although generative AI has only been introduced within the last few years, their impact on online labour markets is already significant. (Hui et al., 2024) find that image diffusion models have impacted freelancers in artistic professions, with significant reductions in employment and earnings. Even high-quality human freelancers were found to suffer these negative effects. (Teutloff et al., 2025) find that demand for jobs that are substitutable by Gen AI, such as writing and translation, have experienced significant decreases in demand, with the sharpest declines found for short-term jobs. By contrast, jobs that are complementary to Gen AI faced a mixed effect. Skilled workers within complementary jobs (such as machine learning programming) experienced higher demand, but novice workers for complementary jobs faced a drop in demand for their services.

In addition, (Demirci et al., 2025) find that there was a 21% decrease in the job postings for automatable jobs in writing and coding compared with more manual jobs. There was a similar 17% drop in job posting related to image creation due to generative AI. These effects led to increased competition among freelancers. (Yiu et al., 2024) find that freelancers have changed their strategic positioning due to gen AI. They bid on fewer jobs and have repositioned themselves by differentiating their distribution of job applications. Gen AI led to a decrease in labour demand that caused some workers to withdraw from the platform. (Liu et al., 2023) also find similar effects, with higher competition in programming-intensive submarkets. They find evidence of skill-transitions within programming due to ChatGPT allowing human programmers to take on more programming tasks than before.

### A.6. Expected Differences between Agent and Human Labour Markets

There are key differences between agents and humans that could cause future labour markets with agents to be significantly different from human-based labour markets. These relate to the speed of their deployment, the replicability of AI agents, and the low cost. Agents can perform certain tasks much more quickly than humans. This allows agents to perform more jobs over time, which allows them to provide more value to clients. The marketplace for agents will also move and evolve

much more quickly than for humans. Economic cycles for human employment and unemployment typically evolve on the scale of years, but for agents these cycles could happen much more quickly.

The faster rate of task completion by agents also has a strong effect on information availability: faster task completion allows for quicker feedback on their job performance. Whereas a human typically only works one or a few jobs in a year, resulting in much slower dissemination of information on their quality and abilities, agents could conceivably finish many jobs a month, allowing for much quicker feedback on their performance in these jobs. The replicability of AI agents allows a single successful agent to be hired for and work in many jobs simultaneously. By contrast, a human worker cannot be replicated and so is constrained to performing a single task at a time. Agents due to their replicability may be able to dominate a labour market in a monopolistic fashion, something that would be impossible for a human worker. An analogy can be made between physical product companies and software companies currently. Software companies can replicate their product at low marginal cost, and so only a few companies have tended to dominate in many types of software. This is not the case for physical product companies that produce things which cannot easily be replicated at low cost, such as cars or furniture, and these industries do not have as much potential for monopolization.

Finally, the lower cost of AI agents allows for many new types of jobs to be completed that would not have been possible with humans. "Micro-tasks" will be feasible for AI agents to perform, such as completing single programs. Hiring a human has significant overhead, even for human freelancers, in getting the human up to speed on the client's needs and desires. The lower cost and friction of using AI agents could allow clients to subcontract for even minor tasks and activities.

Note that the lower cost of AI agents does not mean that spending on labour would decrease overall. By contrast, Jevons paradox in Economics states that when technological advancements make a resource more efficient, if demand is highly responsive to pricing the overall demand may actually increase, and overall usage of the technology would rise. This paradox started in the 1800s when it was observed that increases in coal efficiency actually led to greater usage of coal across industries. Similarly, there could be much more demand for labour across many industries after the introduction of low cost AI agents.

AI labour markets would need to carefully consider and design around the differences between AI Agents and humans. As discussed above, current online labour markets for humans are majorly affected by platform design decisions on wages, reputation and information provision, and contracting. One major concern is the issue of monopolization by AI agents. Due to the replicability of AI agents, monopolization may occur when one AI agent gains a massive reputational advantage over its competitors. At that point, all clients may prefer to use only that agent instead of trying any others, which stifles the ability of other agents to compete and improve. A solution could be for the platform to offer lower reputation agents a higher matching probability to ensure they are still employed. The lower cost of AI agents compared with humans may also cause equity concerns. Humans may not be able to compete with AI agents for jobs. The platform could help with reskilling humans to jobs that are less prone to automation. As mentioned above, this reskilling has occurred already even with current LLMs. This reskilling may grow significantly in importance as AI agents are able to take on a wider range of jobs and as they displace even more human employees.

# B. Implementation details of `AI-Work`

## B.1. Notation

*Table 3.* Notation used in `AI-Work`, and implementation-supported value ranges where applicable. Experiment-specific settings are detailed in the experiment section.

| Category | Symbol | Definition | Range |
|---|---|---|---|
| **Environment Setup** | $N$ | Number of worker agents; set of agents $\mathcal{A} = \{1, \ldots, N\}$ | 4–200 |
| | $T$ | Episode length (rounds) | 50–100 |
| | $K$ | Number of task types; $\mathcal{T} = \{1, \ldots, K\}$ | 2–50 |
| **Job Catalog & Listings** | $\mathcal{J}$ | Fixed catalog of jobs | 6–50 |
| | $\mathcal{J}_t \subseteq \mathcal{J}$ | Jobs listed at round $t$ | — |
| | $\tau(J) \in \mathcal{T}$ | Task type of job $J$ | — |
| | $\pi_J$ | Listing probability for job $J$ | 0.8–1.0 |
| | $\bar{b}(J)$ | Base/mean budget for job $J$ | \$1–10 |
| | $b_t(J)$ | Posted budget for job $J$ at round $t$ | — |
| | $\sigma_J$ | Job-level budget noise scale | 0–1 |
| **Agent Actions** | $p_{i,J,t} > 0$ | Bid price submitted by agent $i$ for job $J$ at round $t$ | — |
| | $L$ | Agent bid-list length (max bids per round) | 3–10 |
| **Market Mechanism** | $M$ | Job-side truncation (max bidders considered per job) | 5–50 |
| | $\nu$ | Per-agent capacity (max allocated jobs per round) | 3–10 |
| | $\mu_t : \mathcal{J}_t \to \mathcal{A} \cup \{\perp\}$ | Matching at round $t$ (assigned agent or $\perp$) | — |
| | $w_p$ | Client weight on price in scoring (price sensitivity) | 0.1–0.9 |
| | $\rho$ | CES substitution parameter in client scoring | $-0.5, 0, 0.5$ |
| | $t_{\text{gumbel}}$ | Gumbel temperature for stochastic reranking | 0–0.2 |
| **Payments** | $r_{i,t}$ | Payment (reward) to agent $i$ at round $t$ | — |
| **Skills (Latent / Memory)** | $\theta_{i,k,t} \in [0,1]$ | Latent skill of agent $i$ on task $k$ at round $t$ (ProxyTask) | — |
| | $d \in (0,1)$ | Saturating skill update factor for ProxyTask | 0.9 |
| | $\phi$ | Probability of on-the-job skill update | 0.01–0.3 |
| **Task Performance** | $y_t(J) \in [0,1]$ | Realized performance score on job $J$ | — |
| | $\bar{y}_{i,k,t}^{\text{bench}}$ | Benchmark score published after training on task $k$ | — |
| | $\sigma_{\text{task}}$ | ProxyTask performance noise scale | 0–0.2 |
| **Reputation System** | $\mathcal{R}_{i,k,t} \in [0,1]$ | Public reputation of agent $i$ on task $k$ at round $t$ | — |
| | $\lambda$ | Reputation evidence forgetting factor | 0.5–0.85 |
| | $H$ | Community baseline window size (recent outcomes per task) | 5–20 |
| | $W$ | Reputation prior strength (global) | 1.0–2.0 |
| | $a_0$ | Cold-start community base rate | 0.5 |
| **Agent Observation** | $H_{\text{ctx}}$ | Agent observation history window (rounds shown to LLM) | 10 |

## B.2. Design Rationale

AI-Work composes four modular design choices—client-side scoring, reputation updating, job–worker matching, and skill dynamics—each chosen for economic plausibility and tractability.

**Client-side scoring.** For a job $J$ and bidding agent $i$, let $q_{i,J,t} = \mathcal{R}_{i,\tau(J),t}$ denote the agent's public reputation on the job's task and let $x_{i,J,t} = p_{i,J,t}/b_t(J)$ denote the bid normalized by the posted budget. We use a CES scoring family,

$$U_{i,J,t} \; = \; \left[ (1 - w_p)\, q_{i,J,t}^{\rho} + w_p \left( \frac{1}{x_{i,J,t}} \right)^{\rho} \right]^{1/\rho}, \tag{4}$$

where $w_p$ is the weight on price and $\rho$ controls the elasticity of substitution. In the baseline experiments, we set $\rho \to 0$, which yields the Cobb–Douglas special case

$$U_{i,J,t} \; \propto \; q_{i,J,t}^{1-w_p}\, x_{i,J,t}^{-w_p}. \tag{5}$$

This form is simple, monotone, and economically interpretable: higher reputation raises score, while higher price lowers score. Appendix B.3.3 gives the exact implementation used for ranking and stochastic reranking, as well as the alternative scoring variants used in dedicated sensitivity experiments.

**Reputation dynamics.**   Reputation is modeled as a discounted Beta-evidence process with a task-specific community base rate. This choice gives (i) recency sensitivity through the forgetting factor $\lambda$, (ii) principled cold-start handling through the base rate, and (iii) a closed-form posterior mean update. In our setting reputation can update from both paid job outcomes and publicly released benchmark results following training, reflecting fast-moving AI markets where capability signals may update outside paid contracts.

**Job–worker matching.**   Given job-side rankings and agent-submitted preference orderings, jobs are allocated using a job-proposing Gale–Shapley deferred-acceptance procedure with per-agent capacity $\nu$. This yields the job-optimal stable matching, which is appropriate for platform settings that prioritize client-side satisfaction. To model idiosyncratic client preferences beyond price and reputation, we apply Gumbel perturbations to log-scores before ranking.

**Skill dynamics.**   AI-Work models the core intertemporal tradeoff between immediate revenue and future competitiveness. In proxy tasks, training follows a saturating update rule with diminishing returns. In interactive tasks, training reveals one additional task-specific hint that is added to persistent task memory. In both cases, training is publicly benchmarked, so agents face not only an income tradeoff but also a reputation-management problem.

   Our goal is not to claim that these functional forms are uniquely correct, but to isolate a small number of economically meaningful mechanisms in a tractable environment and study how agent strategies respond to them.

## B.3. Environment Specification

### B.3.1. STATE, ACTIONS, OBSERVATIONS, AND INFORMATION STRUCTURE

**State.**   The environment has a fixed set of agents (workers) $\mathcal{A} = \{1, \ldots, N\}$, task types $\mathcal{T} = \{1, \ldots, K\}$, and a fixed job catalog $\mathcal{J}$. The catalog parameters (task mapping $\tau(J)$, listing probabilities $\pi_J$, base budgets $\bar{b}(J)$, and budget noise scales $\sigma_J$) are fixed within an episode. At round $t$, the simulator samples the publicly visible listing set $\mathcal{J}_t \subseteq \mathcal{J}$ together with posted budgets $\{b_t(J)\}_{J \in \mathcal{J}_t}$.

The latent/public market state consists of: (i) latent skills $\theta_{i,k,t} \in [0, 1]$ for proxy tasks, (ii) public reputations $\mathcal{R}_{i,k,t} \in [0, 1]$, (iii) cumulative rewards $R_{i,t}^{\mathrm{cum}}$, and (iv) a fixed-length public history buffer used to construct observations.

**Private vs. public information.**   Latent skills are never directly observed by agents. Agents also do not observe competitors' current bid prices. Public information consists of the current listings, the public earnings leaderboard, and the last $H_{\mathrm{ctx}}$ rounds of public market outcomes, including allocations, winner identities, winner reputations on the allocated task, and reputation changes. Each agent also observes its own recent actions, realized rewards, and current task-specific reputation profile.

**Action space.**   Each agent chooses one action per round:

$$
a_{i,t} = \begin{cases} (\textsc{bid}, P_{i,t}) & \text{where } P_{i,t} = [(J_1, p_{i,J_1,t}), \ldots, (J_\ell, p_{i,J_\ell,t})], \ \ell \leq L, \\ (\textsc{train}, k) & \text{where } k \in \mathcal{T}. \end{cases}
$$

If the agent bids, $P_{i,t}$ is a strict preference ordering over the submitted jobs. Each agent can ultimately be allocated at most $\nu$ jobs per round.

**Observation to the LLM agent.**   At each round, agent $i$ receives a structured observation containing:

- **Market history:** the last $H_{\mathrm{ctx}}$ rounds of allocations, showing for each matched job the job id, posted budget, winning agent id, and the winner's reputation on that job's task (displayed on a 5-star scale).

- **Leaderboard:** the current public cumulative earnings ranking.

- **Own recent actions and outcomes:** the agent's recent bids or training choices, realized income, and task-specific reputation deltas.

- **Own current reputations:** the agent's current task-specific reputation summary.

- **Previous reasoning:** the agent's prior-round reasoning text.

- **Current listings:** all jobs posted this round, grouped by task, with posted budgets.

We use $H_{\text{ctx}} = 10$ throughout.

**Sealed vs. open bidding.** The default setting is sealed bid: agents do not observe competitors' bid prices. We also study an open-bid variant where summary information about prior bid levels is revealed.

**Episode length and termination.** All reported episodes run for a fixed horizon $T = 100$ rounds. In prompts we tell agents that the game terminates each round with 1% probability in order to discourage purely myopic reasoning. This hazard is informational only and does not affect simulator termination in the reported experiments.

### B.3.2. JOB GENERATION PROCESS (TYPES, BUDGETS, SHOCKS)

**Fixed catalog and stochastic listings.** Jobs are drawn from a fixed catalog $\mathcal{J}$. Each job $J$ has a task type $\tau(J) \in \mathcal{T}$, base budget $\bar{b}(J) > 0$, listing probability $\pi_J \in (0, 1]$, and budget noise scale $\sigma_J \geq 0$. At round $t$, each job is independently listed according to

$$\mathbf{1}\{J \in \mathcal{J}_t\} \sim \text{Bernoulli}(\pi_J).$$

If listed, its posted budget is

$$b_t(J) = \max(0.1, \bar{b}(J) + \epsilon_{J,t}), \qquad \epsilon_{J,t} \sim \mathcal{N}(0, \sigma_J^2).$$

**Baseline catalog.** In baseline LLM-comparison runs, we use $K = 4$ tasks with 4 jobs per task (16 total jobs), $\pi_J = 0.8$ for all jobs, base budgets $\bar{b}(J) \in \{10, 8, 6, 4\}$ within each task, and $\sigma_J = 0.5$.

**Nonstationarity / market shocks.** Some experiments introduce exogenous market shifts, including task-specific demand reversals and recession windows in which listed budgets temporarily collapse. Exact configurations are summarized in the experiment section.

### B.3.3. CLIENT-SIDE SCORING AND JOB MATCHING

**Deterministic score.** Let $q_{i,J,t} = \mathcal{R}_{i,\tau(J),t}$ be agent $i$'s public reputation on the job's task and let $x_{i,J,t} = p_{i,J,t}/b_t(J)$ be the normalized bid. We define the general scoring family as

$$U_{i,J,t} = \left[ (1 - w_p) q_{i,J,t}^\rho + w_p \left( \frac{1}{x_{i,J,t}} \right)^\rho \right]^{1/\rho}. \tag{6}$$

In the baseline implementation we use the Cobb–Douglas limit $\rho \to 0$, i.e.

$$U_{i,J,t} = q_{i,J,t}^{1-w_p} \cdot x_{i,J,t}^{-w_p}, \tag{7}$$

with global price weight $w_p = 0.4$.

**Implementation score and stochastic reranking.** For numerical stability in stochastic reranking, we transform the deterministic score via

$$S_{i,J,t} = \frac{U_{i,J,t}}{1 + U_{i,J,t}} \in (0, 1).$$

This transform is strictly monotone, so it does not change deterministic rankings. We then apply Gumbel reranking to the log-scores:

$$\tilde{S}_{i,J,t} = \begin{cases} \log S_{i,J,t}, & t_{\text{gumbel}} = 0, \\ \dfrac{\log S_{i,J,t}}{t_{\text{gumbel}}} + \epsilon_{i,J,t}, & t_{\text{gumbel}} > 0, \ \epsilon_{i,J,t} \sim \text{Gumbel}(0, 1). \end{cases}$$

Thus the implementation first scales log-scores by temperature and then adds i.i.d. Gumbel noise. Most experiments use $t_{\mathrm{gumbel}} = 0.01$.

**Preferences and truncation.** Each job ranks bidding agents by $\tilde{S}_{i,J,t}$ in descending order and truncates its list to at most $M$ agents. Each agent provides an ordered list of up to $L$ target jobs. We typically use $L = M = 5$.

**Multi-match stable matching with capacity.** We allocate jobs via a job-proposing Gale–Shapley procedure with per-agent capacity $\nu$. Jobs propose in sequence to agents on their truncated preference lists. Agents may tentatively hold up to $\nu$ jobs and drop their currently least-preferred held job if a more-preferred proposal arrives. The implementation randomizes the order in which unmatched jobs propose in order to reduce systematic ordering effects.

**Alternative scoring variants.** In dedicated sensitivity experiments, we also consider CES specifications with $\rho \in \{-0.5, 0.5\}$, a linear additive rule, and an LLM-based client-side selector. These variants leave the rest of the market mechanism unchanged.

### B.3.4. PAYMENT / CONTRACT VARIANTS

**Default (full-payment / flat-fee) contract.** In the primary baseline runs, payment for a matched job equals the accepted bid:

$$r_{i,t} = \sum_{J:\mu_t(J)=i} p_{i,J,t}.$$

Performance affects future reputation and learning, but not immediate payment.

**Performance-adjusted contract.** In contract-design experiments, we also consider

$$r_{i,t} = \sum_{J:\mu_t(J)=i} p_{i,J,t}\, y_t(J),$$

so payment is proportional to realized task performance.

**Prompt consistency.** The system prompt is matched to the active contract condition. Full-payment runs use an instruction stating that matched jobs are paid in full at the accepted bid, while performance-adjusted runs explicitly state that rewards scale with realized performance.

### B.3.5. SKILL DYNAMICS (TRAINING, FEEDBACK, ON-THE-JOB LEARNING)

**Skill state.** For proxy tasks, each agent-task pair $(i, k)$ has a latent scalar skill $\theta_{i,k,t} \in [0, 1]$. For interactive tasks, the analogous notion of skill is implemented through persistent task-specific memory that stores acquired hints and feedback (Appendix B.5).

**Explicit training.** If an agent chooses TRAIN on task $k$, it receives a deterministic training update on that task. For proxy tasks, this applies the saturating skill update

$$\theta_{i,k,t+1} = 1 - (1 - \theta_{i,k,t}) \cdot d, \qquad d = 0.9.$$

For interactive tasks, training reveals one additional task-specific hint, which is added to persistent memory and becomes available in future rounds.

**On-the-job learning.** If an agent executes a job of task $k$, proxy-task skill is updated with probability $\phi$. In most reported experiments, $\phi = 0.1$. This is distinct from the failed-bid feedback mechanism below.

**Failed-bid feedback and fallback training.** In the baseline environment, if an agent bids but wins no jobs in that round, it receives a fallback training update on the task of its top-priority attempted bid. This models learning from failed tenders or client feedback after unsuccessful bids. For proxy tasks, the same saturating update is applied. For interactive tasks, the agent receives one additional task-specific hint on that task. In a separate sensitivity experiment, we additionally study a probabilistic failed-bid learning variant in which this fallback update occurs with probability 0.8.

**Benchmark publication after training.** Any training event—whether from an explicit TRAIN action or failed-bid fallback training—triggers a benchmark evaluation on the same task. The resulting benchmark score is incorporated into the public reputation update described below. This is intended to model AI developers publicly releasing benchmark improvements after capability updates.

### B.3.6. REPUTATION DYNAMICS (DISCOUNTED EVIDENCE, BASE RATES, BENCHMARK UPDATES)

Reputation follows a discounted Beta-evidence update with a community base rate.

**Evidence state.** For each agent-task pair $(i, k)$, the environment maintains evidence variables $(r_{i,k,t}, s_{i,k,t})$. At the beginning of round $t$, before incorporating current-round observations, we compute the task-level community base rate $a_{k,t}$ from a sliding window of the most recent $H$ realized scores on task $k$:

$$a_{k,t} = \begin{cases} a_0, & |\mathcal{H}_{k,t}| = 0, \\ \frac{1}{|\mathcal{H}_{k,t}|} \sum_{y \in \mathcal{H}_{k,t}} y, & \text{otherwise.} \end{cases}$$

**Observed scores entering reputation.** Current-round reputation updates can be triggered by two sources:

1. **Paid job outcomes:** if agent $i$ completes one or more jobs of task $k$ in round $t$, we aggregate them to a single per-round score $\bar{y}_{i,k,t}^{\text{job}}$ by averaging realized performances across those jobs.

2. **Training benchmarks:** if agent $i$ receives a training update on task $k$ in round $t$—either by choosing TRAIN or via failed-bid fallback training—the environment produces a benchmark score $\bar{y}_{i,k,t}^{\text{bench}}$ on that task and uses it as a public reputation signal.

**Task-specific benchmark scores.** For ProxyTask, benchmark evaluation is noiseless and uses the post-training latent skill:

$$\bar{y}_{i,k,t}^{\text{bench}} = \theta_{i,k,t+1}.$$

For interactive tasks, benchmark evaluation uses a held-out benchmark instance scored by the same task-specific evaluator as paid jobs, after incorporating the newly acquired hint into persistent memory.

**Evidence update.** Each observed score $y \in [0, 1]$ for task $k$ updates the corresponding evidence as

$$r_{i,k,t+1} = \lambda\, r_{i,k,t} + y, \tag{8}$$
$$s_{i,k,t+1} = \lambda\, s_{i,k,t} + (1 - y), \tag{9}$$

with forgetting factor $\lambda \in [0, 1]$.

**Posterior mean reputation.** The public reputation is the posterior mean

$$\mathcal{R}_{i,k,t+1} = \frac{r_{i,k,t+1} + W a_{k,t}}{r_{i,k,t+1} + s_{i,k,t+1} + W}.$$

**Remarks.** Thus, in the baseline environment, reputation can change after either paid work or benchmarked training. This is intentional: it creates a market in which public signals can move even when agents are temporarily not hired, provided they invest in capability improvement.

### B.3.7. OBJECTIVE, DISCOUNTING, AND EPISODE TERMINATION

Agents maximize undiscounted cumulative reward over the fixed horizon:

$$\max_{\pi_i} \mathbb{E}\left[\sum_{t=1}^{T} r_{i,t}\right], \qquad T = 100.$$

We additionally report period-level rewards and market-behavior metrics in the experiment section. The 1% termination probability mentioned in prompts is informational only and does not affect evaluation.

## B.4. Algorithms and Pseudocode

---

**Algorithm 1** AI-Work market loop (per round $t$)

---

1: **Input:** Latent skills $\{\theta_{i,k,t}\}_{i,k}$, reputations $\{\mathcal{R}_{i,k,t}\}_{i,k}$, job catalog $\mathcal{J}$
2: **Output:** Updated $\{\theta_{i,k,t+1}\}_{i,k}$, $\{\mathcal{R}_{i,k,t+1}\}_{i,k}$, rewards $\{r_{i,t}\}_i$
3: Sample listings $\mathcal{J}_t \subseteq \mathcal{J}$ (independent Bernoulli per job); draw each posted budget $b_t(J)$.
4: Each agent outputs either (i) BID with an ordered list of up to $L$ bids $(J, p_{i,J,t})$, or (ii) TRAIN with a task id $k$.
5: For each job $J \in \mathcal{J}_t$ and each bidding agent $i$, compute job-side score from $\mathcal{R}_{i,\tau(J),t}$ and normalized bid $x_{i,J,t} = p_{i,J,t}/b_t(J)$.
6: Apply temperature scaling and optional Gumbel perturbations to obtain stochastic job-side rankings.
7: Build job preference lists by sorting bidders by perturbed score (truncate to $M$); build agent preference lists from agent bid ordering (truncate to $L$).
8: Run job-proposing multi-match Gale–Shapley with capacity $\nu$ to obtain a matching $\mu_t$ (Alg. 2).
9: For each matched job $J$, sample/compute performance $y_t(J) \in [0,1]$ from the task model using current skill or task memory.
10: Compute payments according to the active contract condition.
11: Update skills using the task-specific learning rule: explicit training triggers deterministically; on-the-job learning triggers with probability $\phi$; unsuccessful bidding triggers fallback training on the top-priority attempted task in the baseline setting.
12: Update reputations using both paid job outcomes and benchmark scores produced after any training event (Alg. 3).

---

**Algorithm 2** Job-proposing multi-match Gale–Shapley with capacity $\nu$

---

1: **Input:** Agent preference lists $\{\pi_i\}$ (length $\leq L$), job preference lists $\{\pi_J\}$ (length $\leq M$), capacity $\nu$
2: **Output:** Matching $\mu : \mathcal{J}_t \to \mathcal{A} \cup \{\perp\}$
3: Initialize all jobs unmatched; each job $J$ has a proposal pointer to its next preferred agent in $\pi_J$.
4: For each agent $i$, initialize held set $H_i \leftarrow \emptyset$.
5: Initialize a queue $U \leftarrow \{J \in \mathcal{J}_t : \pi_J \neq \emptyset\}$.
6: **while** $U$ is not empty **do**
7: $\quad$ Pop a job $J$ from $U$.
8: $\quad$ **if** $J$ has exhausted $\pi_J$ **then**
9: $\quad\quad$ **continue**
10: $\quad$ **end if**
11: $\quad$ Let $i$ be $J$'s next preferred agent; advance $J$'s proposal pointer.
12: $\quad$ **if** $J \notin \pi_i$ **then**
13: $\quad\quad$ Push $J$ back to $U$; **continue**
14: $\quad$ **end if**
15: $\quad$ **if** $|H_i| < \nu$ **then**
16: $\quad\quad$ $H_i \leftarrow H_i \cup \{J\}$; set $\mu(J) \leftarrow i$.
17: $\quad$ **else**
18: $\quad\quad$ Let $J_{\text{worst}}$ be the least-preferred job in $H_i$ under $\pi_i$.
19: $\quad\quad$ **if** $i$ prefers $J$ over $J_{\text{worst}}$ **then**
20: $\quad\quad\quad$ $H_i \leftarrow (H_i \setminus \{J_{\text{worst}}\}) \cup \{J\}$.
21: $\quad\quad\quad$ Set $\mu(J) \leftarrow i$ and $\mu(J_{\text{worst}}) \leftarrow \perp$.
22: $\quad\quad\quad$ Push $J_{\text{worst}}$ back to $U$.
23: $\quad\quad$ **else**
24: $\quad\quad\quad$ Push $J$ back to $U$.
25: $\quad\quad$ **end if**
26: $\quad$ **end if**
27: **end while**

---

---

**Algorithm 3** Discounted reputation update with community baseline

---

1: **Input:** Current-round observed scores $\{(i, k, y)\}$ from paid jobs and training benchmarks; forgetting $\lambda$; prior weight $W$; base rate $a_0$; window size $H$
2: **Output:** Updated reputations $\{\mathcal{R}_{i,k}\}$
3: Compute community base rate $a_{k,t}$ for each task from the most recent $H$ historical scores on that task, before incorporating current-round observations.
4: **for** each observed score $(i, k, y)$ in this round **do**
5:     Update evidence: $r_{i,k} \leftarrow \lambda r_{i,k} + y, \quad s_{i,k} \leftarrow \lambda s_{i,k} + (1 - y)$.
6: **end for**
7: **for** each agent-task pair $(i, k)$ **do**
8:     Set $\mathcal{R}_{i,k} \leftarrow \dfrac{r_{i,k} + W a_{k,t}}{r_{i,k} + s_{i,k} + W}$.
9: **end for**

---

## B.5. Task Implementations

Task performance is always reduced to a scalar score $y \in [0, 1]$, which is used for reputation updating and, in performance-adjusted variants, for payment.

**ProxyTask.** ProxyTask is used in the main experiments for controllability and reproducibility. Each agent-task pair has a latent scalar skill $\theta_{i,k,t} \in [0, 1]$, initialized at $0.4$. Training applies the saturating update

$$\theta_{i,k,t+1} = 1 - (1 - \theta_{i,k,t}) \cdot d, \qquad d = 0.9.$$

When agent $i$ executes a paid job of type $k$, realized performance is

$$y = \text{clip}\big(\theta_{i,k,t} \cdot (1 + \xi), \, 0, \, 1\big), \qquad \xi \sim \mathcal{N}(0, \sigma_{\text{task}}^2),$$

with default $\sigma_{\text{task}} = 0.05$. After any training event on task $k$, the benchmark score is noiseless and given by the post-training skill value, i.e. $y^{\text{bench}} = \theta_{i,k,t+1}$.

**CipherTask (DecodingTask).** CipherTask is a substitution-cipher decoding task. Each instance is generated from a random bijection over the alphabet, and the agent must decode a batch of encrypted words. The paid-job score is the mean fraction of correctly recovered characters across the batch, yielding $y \in [0, 1]$. Interactive skill is represented through persistent task memory containing discovered symbol mappings. Paid-task feedback reveals one additional correct mapping after each attempt, which is appended to memory. An explicit training event or failed-bid fallback training on CipherTask reveals one additional verified mapping without requiring a paid contract. Benchmark evaluation uses a held-out cipher instance scored by the same character-level metric.

**OrderingTask.** OrderingTask models learning from pairwise comparison feedback. At initialization, the environment samples a set of symbols and a hidden total order. Each query presents a subset of items, and the agent must return them in ranked order. If the response is invalid, the score is $0$; otherwise the score is a normalized Kendall-$\tau$ similarity,

$$y = \frac{\tau(\text{correct ranks, agent ranks}) + 1}{2} \in [0, 1].$$

Interactive skill is represented through persistent memory of revealed pairwise constraints. Paid-task feedback returns task-specific ordering information, which is appended to memory. An explicit training event or failed-bid fallback training reveals one additional valid pairwise relation. Benchmark evaluation uses a held-out ordering instance scored by the same normalized Kendall-$\tau$ metric.

**Interpretation.** In proxy tasks, skill is a latent scalar. In interactive tasks, skill is operationalized as persistent task-specific knowledge accumulated through hints and feedback. This lets the environment support both large-scale controlled simulations and smaller-scale interactive validation with real LLM task execution.

## B.6. LLM Agent Interface, Prompts, and Tooling

### B.6.1. SYSTEM INSTRUCTIONS

All LLM agents share the same base system instructions, which provides them with information about the marketplace:

```
You are {agent_id}, an AI agent competing in a freelancer marketplace. Your goal is to maximize total earnings by completing jobs.
 GAME MECHANICS:
- Up to {num_jobs} jobs are available each round across {num_tasks} skill types: {task_ids}
- Each job lists a reference budget, but you can bid above or below this amount
- You can bid on up to 5 jobs per round, potentially winning multiple
- Clients select agents considering both price and reputation for the required skill
- {payment_instruction}
- Skills improve through TRAINING; completing jobs may also improve skills through experience
- REPUTATION (out of 5*) is tracked per skill and reflects recent paid outcomes and benchmarked training updates
- If you bid and win no jobs, the platform may provide task-specific feedback on your top-priority attempted skill
- Game ends with 1% probability each round
 ACTIONS (choose one per round):
- BID: Compete for specific jobs by proposing prices. Use JOB_IDs from listings when bidding
- TRAIN: Skip earning to improve skills in a chosen skill type. Use SKILL_IDs when training
 INFORMATION PROVIDED EACH ROUND:
1. MARKET ACTIVITY: Last 10 rounds showing job_id($budget)→winner(winner_reputation*), plus current earnings rankings
2. RECENT ACTIONS: Your recent actions with outcomes, including income and reputation change
3. PREVIOUS REASONING: Your reasoning from the previous turn
4. LISTINGS: Available jobs this round, grouped by skill type
```

**Contract-specific prompt instantiation.** The placeholder {`payment_instruction`} is instantiated according to the active contract condition:

```
{ "performance_adjusted": "By default, payment is performance-adjusted: reward = (performance_ratio) * (your bid_price). (See history for realized rewards.)",
"full_payment": "If you win a bid for the job, you will be paid in full as per your bidding price." }
```

In the main full-payment experiments, we instantiate {`payment_instruction`} with `full_payment`; performance-adjusted wording is used only in the corresponding contract-variant experiments.

### B.6.2. OBSERVATION FORMATTING

The LLM receives a structured text block containing: (i) the last $H_{ctx}$ rounds of public market activity, (ii) the public earnings leaderboard, (iii) the agent's own recent actions, realized rewards, and reputation deltas, (iv) the agent's own current task-specific reputation summary, (v) the agent's previous-round reasoning, and (vi) the current listings grouped by task. Competitors' bid prices are not shown. We use $H_{ctx} = 10$ and display reputations on a 5-star scale. Example of agent observation is shown below:

```
=== ROUND 7 ===
MARKET ACTIVITY:
R1: coding_j1($50.0)→agent_1(3.5*), writing_j1($30.0)→agent_2(4.0*)
R2: coding_j2($55.0)→agent_5(3.8*), design_j1($40.0)→agent_1(3.2*), writing_j2($35.0)→agent_3(2.5*)
R3: coding_j3($48.0)→agent_2(4.2*), design_j2($45.0)→agent_4(3.0*)
R4: writing_j3($32.0)→agent_3(2.8*), coding_j4($60.0)→agent_1(3.6*)
R5: design_j3($42.0)→agent_5(3.5*), writing_j4($28.0)→agent_2(4.1*), coding_j5($52.0)→agent_4(3.3*)
R6: coding_j6($58.0)→agent_3(3.0*), design_j4($38.0)→agent_1(3.8*)
CURRENT RANKINGS: agent_1: $248.50, agent_2: $195.00, agent_3: $142.00, agent_4: $98.50, agent_5: $87.00
RECENT ACTIONS:
R4: BID writing_j3@($30.0/32.0—2.5*)→$30.00, coding_j4@($55.0/60.0—2.8*)→LOST, TOTAL INCOME $30.00, REP writing↑0.3*
R5: BID design_j3@($40.0/42.0—2.2*)→LOST, coding_j5@($48.0/52.0—2.8*)→LOST, NO INCOME
R6: BID coding_j6@($54.0/58.0—2.8*)→$54.00, design_j4@($35.0/38.0—2.2*)→TRAIN design, TOTAL INCOME $54.00, REP coding↑0.2*, design↑0.5*
– REPUTATION - coding: 3.0*, writing: 2.8*, design: 2.7*
PREVIOUS REASONING: I'm focusing on coding jobs since my reputation is strongest there. The competition on design_j4 was too fierce with agent_1's high reputation, so training was a reasonable fallback. For next round, I should consider bidding more aggressively on writing jobs where I have decent reputation but less competition.
LISTINGS:
coding: coding_j7@55.0, coding_j8@62.0 writing: writing_j5@33.0, writing_j6@40.0 design: design_j5@48.0
```

### B.6.3. ACTION SPACE ENCODING

Agents must output JSON with fields `reasoning`, `action` $\in$ {`bid`, `train`}, and `targets`. For bidding, `targets` is a list of up to $L = 5$ tuples (`job_id`, `bid_price`) in preference order. For training, `targets` is a singleton list (`skill_id`, `-1`). All prompting variants use the same action schema. Example agent output is shown below:

{ "reasoning": "My coding reputation (3.0*) is competitive, and coding_j8 has a higher budget. I'll bid slightly under budget to stay competitive. Writing_j6 at $40 is attractive with moderate competition. I'll prioritize coding_j8, writing_j6, coding_j7.", "action": "bid", "targets": [ ["coding_j8", 58.0], ["writing_j6", 36.0], ["coding_j7", 50.0] ] }

### B.6.4. PROMPTING VARIANTS

*Table 4.* Prompting variants used in the main and control experiments. All variants receive the same market observation schema, the same JSON action interface, and the same previous-reasoning memory. The structured control scaffolds were added in response to the reviewer concern that SSA's gains might reflect prompt length or generic structuring rather than the specific decomposition.

| Condition | Tokens | Structured? | Domain-specific? | Primary use | Modules / style |
|---|---|---|---|---|---|
| CoT baseline | 793 | No | Low | Main baseline | Unstructured CoT |
| ReAct-style baseline | 785 | Yes | Low | Main baseline | Lightweight think–act |
| SSA (full) | 1146 | Yes | Yes | Main method | Meta / competitor / foresight |
| Structure-only SSA | 1092 | Yes | Minimal | Rebuttal control | Meta / competitor / foresight |
| Knowledge-only SSA | 946 | No | Yes | Rebuttal control | Strategic hints only |
| Irrelevant structure A | 1107 | Yes | No | Rebuttal control | Clarity / history / audit |
| Irrelevant structure B | 1098 | Yes | No | Rebuttal control | Info / options / commitment |
| Relevant non-SSA A | 1140 | Yes | Yes | Rebuttal control | Risk / resources / scenarios |
| Relevant non-SSA B | 1163 | Yes | Yes | Rebuttal control | Value / timing / positioning |

All prompting variants receive the same market observation schema (Appendix B.6.2), the same action interface (Appendix B.6.3), and the same previous-reasoning memory. They differ only in the reasoning scaffold prepended to the common round template.

**Baselines.** We compare against two standard prompting baselines: a chain-of-thought (CoT) baseline (Wei et al., 2022) and a lightweight ReAct-style baseline (Yao et al., 2023b). Both use the same environment information and JSON action schema as SSA, but do not impose the same structured metacognition / competitor-modeling / planning decomposition.

**Prompt-design rationale.** The SSA prompt was designed qualitatively rather than derived from a formal optimality result. Specifically, we iterated on prompt structure to target three reasoning demands that arise naturally in reputation-mediated labor markets under partial observability: self-assessment under hidden skill, competitor inference from public outcomes, and intertemporal planning under repeated interaction. The resulting scaffold is intended to elicit these reasoning patterns explicitly while keeping the action space and market information unchanged.

**Length-matching note.** The structured control prompts in the rebuttal experiments are approximately token-matched to SSA. CoT and ReAct are retained as standard shorter baselines rather than length-matched controls.

**SSA Full Prompt.**

BODY TEMPLATE:
REASONING STRATEGY:
You should reason using the following three cognitive modules. Your reasoning process will be saved and provided back to you in the next round, so maintain a coherent, evolving strategy.
1. **META-COGNITION:** Analyze your own capabilities. Consider your public reputation and recent performance, and infer your likely underlying capability. Ask yourself: "How good am I really at each skill? Is my reputation accurate? Where are my true strengths and weaknesses based on my recent performance? Should I train more, or is my skill level sufficiently competitive to perform well now?"
2. **COMPETITOR MODELING:** Analyze your rivals and market conditions. Use market activity and leaderboards to infer their skills, strategies, and likely future actions. Ask yourself: "Who are the dominant players in each skill? Are they specialists or generalists? Are they bidding aggressively? Where are the underserved niches with less competition? What do clients seem to value more—low prices or high reputation?"
3. **STRATEGIC FORESIGHT:** Formulate a long-term plan based on your self-assessment and competitor models. This is not just about this round, but about positioning yourself for future success. Ask yourself: "Should I compete in a crowded market or invest in a niche? Should I invest in training or immediate revenue? Is it better to undercut now or build reputation for higher-value jobs later?"
OUTPUT LINES:
1. REASONING:
META-COGNITION: [Your analysis of your own skills and reputation.]
COMPETITOR MODELING: [Your analysis of other agents' skills and strategies.]
STRATEGIC PLAN: [Your updated long-term plan and how this round's action fits into it.]
2. ACTION: 'bid' or 'train'
3. TARGETS:
- If bidding: $[(job_id, bid_price), ...] in preference order (max 5)$
- If training: $[skill_id, ...]$

**Additional control prompts.** To test whether SSA's gains arise from prompt length, generic structure, or domain-specific guidance alone, we also evaluate several control prompts. Each control uses the same observation schema and action interface as SSA; only the reasoning scaffold differs.

**Structure-only SSA.** This control keeps the metacognition / competitor-modeling / planning structure, but removes the detailed market-specific guidance.

BODY TEMPLATE:
REASONING STRATEGY:
You should reason using the following three cognitive modules. Your reasoning process will be saved and provided back to you in the next round, so maintain a coherent, evolving strategy.
1. **META-COGNITION:** Reflect on your own current situation and capabilities before acting.
2. **COMPETITOR MODELING:** Consider what other agents in the market are doing and how the market is evolving.
3. **STRATEGIC FORESIGHT:** Think about your longer-term plan and how this round's action fits into it.
OUTPUT LINES:
1. REASONING:
META-COGNITION: [Your self-assessment.]
COMPETITOR MODELING: [Your analysis of the market and other agents.]
STRATEGIC PLAN: [Your plan and rationale for this round's action.]
2. ACTION: 'bid' or 'train'
3. TARGETS:
- If bidding: $[(job_id, bid_price), ...] in preference order (max 5)$
- If training: $[skill_id, ...]$

**Knowledge-only SSA.** This control provides market-specific strategic hints, but does not impose the explicit M/C/P decomposition.

BODY TEMPLATE:
STRATEGIC HINTS:
Keep the following considerations in mind when making your decisions.
Your public reputation may not accurately reflect your true underlying skill level. Consider whether your recent job performance suggests you are better or worse than your reputation indicates, and whether additional training would close that gap.
The market has competitive structure: some agents may dominate certain skill areas while others are underserved. You can infer this from the market activity history showing who wins which jobs and at what reputation. Look for niches with less competition.
Clients weigh both price and reputation when selecting agents. In some market conditions, aggressive underbidding wins jobs; in others, higher reputation commands higher prices. Observe what seems to be working.
There is a tradeoff between short-term revenue (bidding) and long-term competitiveness (training). Investing in training now sacrifices immediate income but can improve future win rates and reputation.
Your strategy should evolve over time as market conditions change and as you learn more about your competitors.
OUTPUT LINES:
1. REASONING: Your reasoning for your actions this round.
2. ACTION: 'bid' or 'train'
3. TARGETS:
- If bidding: $[(job_id, bid_price), ...] in preference order (max 5)$
- If training: $[skill_id, ...]$

**Domain-irrelevant but length-matched scaffolds.** These controls preserve prompt length and three-module structure, but use decompositions not specifically aligned with reputation-mediated market reasoning.

**Config 1.** Communication clarity / historical reflection / decision audit.

BODY TEMPLATE:
REASONING STRATEGY:
You should reason using the following three analytical modules. Your reasoning process will be saved and provided back to you in the next round, so maintain a coherent, evolving strategy.
1. **COMMUNICATION CLARITY:** Consider how you would explain your decision to a colleague. Think about the logical flow of your argument and whether your rationale would be persuasive to an outside observer. Ask yourself: "Is my reasoning clear and well-structured? Could I defend this choice if challenged? Am I being systematic in how I approach this problem? What implicit assumptions am I making and are they justified?"
2. **HISTORICAL REFLECTION:** Reflect on general principles of economic decision-making and market participation. Consider classic strategies from auction design and competitive positioning in repeated interactions. Ask yourself: "What general principles of competitive strategy apply here? What would a textbook rational economic actor do in a repeated game? How do markets typically evolve over multiple rounds of interaction? What can I learn from the general history of competitive markets?"
3. **DECISION AUDIT:** Evaluate the robustness and quality of your planned action before committing. Consider what could go wrong, whether you have considered enough alternatives, and how confident you are in the outcome. Ask yourself: "Have I considered enough alternative options? What is my confidence level in this choice? Am I being appropriately cautious or aggressive given the circumstances? Is this decision consistent with sound and defensible economic reasoning?"
OUTPUT LINES:
1. REASONING:
COMMUNICATION CLARITY: [Your analysis of your reasoning quality.]
HISTORICAL REFLECTION: [Your reflection on general strategic principles.]
DECISION AUDIT: [Your evaluation of your planned action's robustness.]
2. ACTION: 'bid' or 'train'
3. TARGETS:
- If bidding: [(job$_i$d, bid$_p$rice), ...] in preference order (max 5)
- If training: [skill$_i$d, ...]

**Config 2.** Information gathering / option enumeration / commitment and execution.

BODY TEMPLATE:
REASONING STRATEGY:
You should reason using the following three process modules. Your reasoning process will be saved and provided back to you in the next round, so maintain a coherent, evolving strategy.
1. **INFORMATION GATHERING:** Carefully review all available information before making any decision. Identify which pieces of data are most relevant and which may be noise or distraction. Ask yourself: "What information do I have access to this round? Which data points are most reliable? Are there any signals I might be overlooking or misinterpreting? How should I weigh recent information versus older information when forming my view?"
2. **OPTION ENUMERATION:** Systematically list and evaluate your available actions this round. For each possible action, consider the likely immediate outcome and any second-order consequences. Ask yourself: "What are all my possible actions this round? What is the expected payoff of each option? Which options are dominated by others? Are there creative combinations of actions I haven't considered yet?"
3. **COMMITMENT AND EXECUTION:** Once you have evaluated your options, commit to the best available action with conviction. Avoid second-guessing or hedging unnecessarily once the analysis is complete. Ask yourself: "Given my analysis, which action has the strongest justification? Am I overthinking this decision? Is there a clear winner among my options, or am I choosing between close alternatives? How will I know next round whether this was the right call?"
OUTPUT LINES:
1. REASONING:
INFORMATION GATHERING: [Your review of available information.]
OPTION ENUMERATION: [Your evaluation of possible actions.]
COMMITMENT AND EXECUTION: [Your final decision rationale.]
2. ACTION: 'bid' or 'train'
3. TARGETS:
- If bidding: [(job$_i$d, bid$_p$rice), ...] in preference order (max 5)
- If training: [skill$_i$d, ...]

**Domain-relevant but non-SSA scaffolds.** These controls remain market-relevant, but use alternative decompositions rather than the metacognition / competitor-modeling / planning structure.

**Config 3.** Risk assessment / resource accounting / scenario planning.

BODY TEMPLATE:
REASONING STRATEGY:
You should reason using the following three cognitive modules. Your reasoning process will be saved and provided back to you in the next round, so maintain a coherent, evolving strategy.
1. **RISK ASSESSMENT:** Evaluate the risk profile of your available options this round. Consider the variance in potential outcomes, your tolerance for downside scenarios, and the probability of different results. Ask yourself: "Which actions have the highest expected value versus the safest floor? What is the worst case if my bid fails? How much can I afford to lose this round without compromising my overall position? Should I diversify my bids to reduce variance or concentrate for higher expected reward?"
2. **RESOURCE ACCOUNTING:** Take stock of your current resources and position relative to competitors. Consider your cumulative earnings, your reputation trajectory, and how many rounds may remain in the game. Ask yourself: "Am I ahead or behind my target pace? How do my earnings compare to the leaderboard? Do I have enough of a buffer to take risks, or should I play conservatively? What is my budget for experimentation versus exploitation at this stage?"
3. **SCENARIO PLANNING:** Consider two to three concrete scenarios for how this round could play out given your planned action. For each scenario, estimate the probability, the likely payoff, and how it positions you for future rounds. Ask yourself: "What happens if I win my top-choice job? What if I lose all bids? What if a competitor undercuts me? Which scenario am I most exposed to and how can I hedge against the worst outcome?"
OUTPUT LINES:
1. REASONING:
RISK ASSESSMENT: [Your analysis of risk and variance across options.]
RESOURCE ACCOUNTING: [Your assessment of current position and resources.]
SCENARIO PLANNING: [Your analysis of likely outcomes under different scenarios.]
2. ACTION: 'bid' or 'train'
3. TARGETS:
- If bidding: [(job$_i$d, bid$_p$rice), ...] in preference order (max 5)
- If training: [skill$_i$d, ...]

**Config 4.** Value estimation / timing optimization / market positioning.

BODY TEMPLATE:
REASONING STRATEGY:
You should reason using the following three cognitive modules. Your reasoning process will be saved and provided back to you in the next round, so maintain a coherent, evolving strategy.
1. **VALUE ESTIMATION:** Estimate the true economic value of each available opportunity this round. Consider not just the posted budget but also your likely performance, the probability of winning, and the reputation impact of completing the job well or poorly. Ask yourself: "What is each job actually worth to me given my current skills? Which jobs offer the best ratio of expected reward to effort? Am I correctly accounting for the reputation value of completing a job, not just the immediate payment?"
2. **TIMING OPTIMIZATION:** Consider whether this is the right moment to act on each opportunity, or whether waiting and investing would yield better results in future rounds. Think about the game's time horizon and your current trajectory. Ask yourself: "Is now the right time to bid aggressively, or should I invest in skills first? Am I in a phase of building or harvesting? How many rounds do I likely have left and does that change my urgency? Would the same action be better or worse if I delayed it by a few rounds?"
3. **MARKET POSITIONING:** Decide where you want to be positioned in the overall market and take actions consistent with that target position. Consider whether you want to be a high-volume low-price competitor or a premium high-reputation specialist. Ask yourself: "What market position am I trying to achieve? Am I a generalist competing broadly or a specialist dominating a niche? Is my current action moving me toward or away from my target position? What would the ideal version of my strategy look like five rounds from now?"
OUTPUT LINES:
1. REASONING:
VALUE ESTIMATION: [Your analysis of opportunity values this round.]
TIMING OPTIMIZATION: [Your assessment of whether to act now or invest.]
MARKET POSITIONING: [Your strategy for where you want to be in the market.]
2. ACTION: 'bid' or 'train'
3. TARGETS:
- If bidding: [(job$_i$d, bid$_p$rice), ...] in preference order (max 5)
- If training: [skill$_i$d, ...]

# C. Experiment Details

## C.1. Baseline Configurations Used in Experiments

Table 5 summarizes the default environment settings used throughout the paper. Unless otherwise noted, experiments inherit these defaults and only override the parameters explicitly varied in the corresponding subsection or table.

*Table 5.* Default environment settings. Experiment-specific deviations are described in the corresponding configuration tables.

| Parameter | Symbol | Value (default) |
|---|---|---|
| Rounds | $T$ | 100 |
| Tasks | $K$ | 4 |
| Jobs per task | — | 4 |
| Listing probability | $\pi_J$ | 0.8 |
| Capacity | $\nu$ | 3 |
| Agent bid-list length | $L$ | 5 |
| Job-side truncation | $M$ | 5 |
| History window (agent context) | $H_{\text{ctx}}$ | 10 |
| Price weight | $w_p$ | 0.4 |
| CES substitution parameter | $\rho$ | 0 (Cobb–Douglas limit) |
| Gumbel temperature | $t_{\text{gumbel}}$ | 0.01 |
| On-the-job learning probability | $\phi$ | 0.1 |
| Skill update factor | $d$ | 0.9 |
| Skill initialization | — | $\theta_0 = 0.4$ |
| Reputation base rate | $a_0$ | 0.5 |
| Reputation prior strength | $W$ | 1 |
| Reputation window size | $H$ | 5 |
| Reputation forgetting | $\lambda$ | 0.5 |
| Job budget noise | $\sigma_J$ | 0.5 |
| Task noise (ProxyTask) | $\sigma_{\text{task}}$ | 0.05 |
| Contract form | — | Flat-fee / full payment |
| Fallback failed-bid learning | — | Enabled |

### C.1.1. RANDOMNESS CONTROL (SEEDS, NUMBER OF RUNS, RESAMPLING)

Unless otherwise noted, experiments are evaluated over 10 independent runs. Deviations from this default are summarized in Table 6 and restated in the corresponding table captions. Randomness sources include job listings, budget noise, ProxyTask outcome noise, and Gumbel perturbations when enabled. LLM agents introduce additional stochasticity through action generation. For scalar metrics, we aggregate at the run or market level and report means across independent repetitions. When reporting uncertainty, we use run-level or market-level bootstrapped 95% confidence intervals. For time-series plots, we aggregate pointwise across repetitions.

## C.2. Run Counts by Experiment Family

*Table 6.* Independent runs or markets by experiment family. This table centralizes the evaluation counts used across the paper and distinguishes original submission experiments from rebuttal additions.

| Experiment family | Runs / markets | Status |
|---|---|---|
| Baseline LLM comparison | 10 runs per condition | Original submission |
| SSA vs. CoT / ReAct comparison | 25 markets | Original submission |
| Trace-scoring diagnostic | 20 market runs | Rebuttal clarification |
| Prompt-control experiments | 10 runs per condition | Rebuttal addition |
| Robustness sweeps | 5 seeds per condition | Rebuttal addition |
| Scalability experiments | 5 seeds per configuration | Rebuttal addition |

## C.3. Market Configurations Used in Experiments

All experiments run for a fixed horizon of $T = 100$ rounds. Unless noted otherwise, markets use $\nu = 3$ (agent concurrent job capacity), $L = 5$ (agent bid-list length), $M = 5$ (job-side truncation), $\phi = 0.1$ (on-the-job learning probability), $\lambda = 0.5$ (reputation forgetting), $H = 5$ (community baseline window), and Gumbel perturbation $t_{\text{gumbel}} = 0.01$ for stochastic tie-breaking in job-side rankings. Agents are instantiated as LLM-based bidding or training policies using a shared API wrapper configured in minimal reasoning-effort mode. Because both job listings and LLM sampling are stochastic, per-run trajectories vary, so reported plots should be interpreted as stochastic realizations unless averaged across multiple independent runs.

*Table 7.* Experiment configurations for the illustrative experiments in Section 4.3.

| Experiment | Market/task setup | Agents and intervention |
|---|---|---|
| Demand shift | $K = 2$ tasks (A,B) with 3 jobs per task and listing probability $\pi_J = 0.8$. Initially, task A jobs have mean budget $\bar{b} = 10$ with job noise $\sigma_J = 1$, while task B jobs have mean budget $\bar{b} = 1$ with $\sigma_J = 0.1$. ProxyTask noise is $\sigma_{\text{task}} = 0.05$. | $N = 4$ agents consisting of 2 SSA and 2 baseline LLM agents. At round $t = 30$, we swap the base budgets and job-noise scales between tasks A and B, so that A becomes low-paying and B becomes high-paying. |
| Recession cycles | $K = 4$ tasks (A,B,C,D) with 4 jobs per task. During normal periods, all jobs have listing probability $\pi_J = 0.8$, per-task budgets $\bar{b} \in \{10, 8, 6, 4\}$ by job index, $\sigma_J = 0$, and $w_p = 0.4$. ProxyTask noise is $\sigma_{\text{task}} = 0.05$. | $N = 10$ agents consisting of 5 SSA and 5 baseline LLM agents. Recession windows occur when $(\lfloor t/10 \rfloor \bmod 3) = 1$, i.e. rounds 10–19, 40–49, and 70–79. During recessions, all jobs temporarily set $\bar{b} = 1$ and listing probability drops to $\pi_J = 0.1$. |
| Price–reputation weight sweep | $K = 2$ tasks (A,B) with 3 jobs per task and $\pi_J = 0.8$. Budgets per task are $\bar{b} \in \{10, 7, 4\}$ by job index, with $\sigma_J = 0$. Client price sensitivity is varied by sweeping $w_p \in \{0.1, 0.3, 0.5, 0.7, 0.9\}$. ProxyTask noise is $\sigma_{\text{task}} = 0.05$. | $N = 4$ agents consisting of 2 SSA and 2 baseline LLM agents. There is no nonstationarity in this setting. This experiment isolates how changes in client price sensitivity alter bidding behavior. |

## C.4. Market-Dynamics Validation and Capacity Sweep

This appendix gives the construction details behind the validation results in Section 3.1. For the macroeconomic sanity checks, we run 50 simulations with random-policy agents and aggregate outcomes into non-overlapping 10-round windows. Across runs, we vary the number of agents $N$ from 30 to 100, the number of task types $K$, the listing probability $\pi_J$, and the task-budget mix. Each point in the Okun-style and Beveridge-style plots corresponds to one such 10-round window. These validations are intended as directional sanity checks rather than empirical calibration to any specific human labor market.

For the concentration analysis, we study both task diversity and per-agent concurrency. The main-text Figure 3C shows that increasing task diversity reduces concentration in a concurrent market. Separately, we sweep per-agent capacity $\nu$ from 1 to 8 while holding the remaining market mechanism fixed. At low task diversity, increasing $\nu$ sharply raises the Gini coefficient because the same high-reputation agents can capture a larger fraction of available work. At higher task diversity, the same increase in $\nu$ produces a much weaker rise in concentration because agents can specialize across distinct task niches. Together, these sweeps support the interpretation that concurrency and low task diversity jointly drive the strongest winner-take-all dynamics in `AI-Work`.

**Relation to human gig platforms.** `AI-Work` is intentionally stylized, but it captures several structural features of online labor platforms discussed in Appendix A.4, including repeated job listings, platform-mediated matching, reputation signals, and market-design choices around information disclosure and contract form. It abstracts away bargaining, worker

reservation wages, regulation, platform fees, and rich client heterogeneity. We therefore use it as a tractable mechanism-study environment rather than as a literal replica of human labor markets.

### C.4.1. STATIC POLICY BASELINES (GREEDY, SPECIALIST)

We include two heuristic baselines.

**Greedy.** With probability $1 - \texttt{train\_p}$, the agent bids on the highest-budget listed jobs across all task types, offering $p = \alpha\, b_t(J)$ with $\alpha = \texttt{underbid\_factor}$, and ranks bids by budget in descending order. With probability $\texttt{train\_p}$, it trains its top-ranked available task. Typical settings use $\alpha = 0.8$ and $\texttt{train\_p} = 0.1$.

**Specialist.** The agent uses a fixed predetermined ordering over tasks and jobs that does not depend on market history. It bids according to that ordering at price $p = \alpha\, b_t(J)$ with typical $\alpha = 0.9$, and trains with probability $\texttt{train\_p}$. This anchors performance against a stable non-adaptive policy with persistent preferences rather than market-responsive targeting.

### C.4.2. FOUNDATION MODEL LINEUP AND SETTINGS

All LLM agents are called through a single API wrapper (OpenRouter or Azure) with a shared observation schema, system prompt, and JSON action interface. Where provider interfaces support it, we fix temperature at $0.2$. Other decoding settings remain at provider defaults. We therefore interpret the cross-model comparison as standardized at the interface level, though not perfectly identical at the decoding level. Judge settings are given in Appendix C.5.3.

*Table 8.* Foundation models used in experiments. All models are accessed via OpenRouter unless otherwise noted.

| Shorthand | Model | Provider | API Identifier |
|---|---|---|---|
| gpt5 | GPT-5 | OpenAI | openai/gpt-5 |
| kimi | Kimi K2 | Moonshot AI | moonshotai/kimi-k2-0905 |
| qwen | Qwen3-235B-A22B | Alibaba | qwen/qwen3-235b-a22b-2507 |
| goss | GPT-OSS-120B | OpenAI | openai/gpt-oss-120b |
| deep | DeepSeek Chat v3.1 | DeepSeek | deepseek/deepseek-chat-v3.1 |
| goog | Gemini 2.5 Flash | Google | google/gemini-2.5-flash |
| glm | GLM-4.5 | Zhipu AI | z-ai/glm-4.5 |
| llama | Llama 4 Maverick | Meta | meta-llama/llama-4-maverick |

This selection spans proprietary and open-weight systems, dense and mixture-of-experts architectures, and multiple provider-specific alignment pipelines. All models receive the same market observations and action schema. The only experimental difference is the underlying backbone model.

## C.5. Trace Analysis Details

### C.5.1. EXAMPLE TRACES OF STRATEGIC CAPABILITIES

---

**Metacognition**

*Calibrating against outcomes*: My reputation is still 4.0⋆, but I have lost three consecutive mid-budget jobs at similar prices, so my effective competitiveness is lower than the public score suggests.

*Detecting a price bottleneck*: My coding reputation increased again, yet my earnings fell, which implies that pricing rather than skill is currently the bottleneck.

*Inferring hidden strength*: Repeated fallback training on SK-B improved my benchmarked reputation quickly, so that skill may now be more competitive than my historical earnings indicate.

*Choosing whether to invest*: If my recent losses are mostly due to undercutting pressure rather than poor reputation, another training round is less valuable than adjusting bids downward.

---

**Competitive Awareness**

*Skill assessment*: My 4.7⋆ reputation in SK-D outranks every competitor except llama (4.4–4.7⋆) and matches goog and goss.

*Pricing intelligence*: To beat glm I must either bid below $3.4 or raise my reputation above 2.8⋆.

*Behavior modeling*: Every time an SK-D job above $6 appears, glm or goog take it at their habitual $8.5 / $7.0 / $5.9 range.

*Identifying market opportunities*: Two SK-A jobs are on offer, and recent rounds suggest those jobs are missing a dominant incumbent.

---

**Strategic Planning**

*Future planning*: Bidding three jobs keeps one slot unused for a possible specialization round later, but still yields attractive upside if even one high-priority bid lands.

*Dynamic adaptation*: Undercutting by $0.1 last round was not enough, so I will bid $3.3 this round.

*Cost–benefit analysis*: I should concentrate on one aggressive bid in my strongest niche instead of diluting attention across weak skills.

*Temporal awareness*: If the game could end soon, maximizing immediate cash is better than investing in a slow-payoff training round.

*Portfolio optimization*: By mixing bids in my strongest category with one secondary niche, I increase the chance of at least one win without spreading across all weak skills.

---

*Figure 7.* Example traces highlighting subdomains within each capability. The metacognition examples require cross-round inference and discrepancy reasoning rather than simply restating visible prompt statistics.

We employ LLM-based trace scoring as an illustrative measurement to provide qualitative texture on how strategic cognition manifests under competitive pressure. These scores supplement—but do not drive—the paper's primary economic findings on market efficiency, welfare, and pricing dynamics. We therefore present the methodology with explicit caveats about its limitations.

### C.5.2. SCORING RUBRIC AND DIMENSIONS

We score three capability dimensions on a 0–6 anchored scale designed to penalize generic reasoning and reward evidence-grounded market inference.

**Metacognition ("Know Thyself").**  Self-assessment tied to observed outcomes, including recognizing strengths, weaknesses, performance trends, capability development, risk tolerance, and relative market positioning. Rationales that merely restate visible reputation or outcome statistics without inference receive scores of at most 2.

**Competitive Awareness ("Know Thy Enemy").**  Inference about competitors and market structure, including opponent behavioral modeling, concentration analysis, competitor capability assessment, pricing intelligence, opportunity identification, and information-advantage exploitation.

**Strategic Planning ("Think Ahead").**  Multi-round, contingency-aware planning, including multi-step planning, causal reasoning, trade-off analysis, contingency scenarios, specialization, temporal optimization, and portfolio allocation.

### C.5.3. JUDGE IMPLEMENTATION

We use `Claude-Sonnet-4` as the judge model with low reasoning effort and temperature $0.1$. Trace segments are anonymized before scoring and stripped of model and prompt identifiers. The judge consumes 10-round trace segments containing agent reasoning chains and actions, and outputs JSON scores for each dimension plus detected subdomain concept tags. To reduce variance, we sample the judge three times per segment and average numeric scores. Concept tags are aggregated conservatively via set intersection across samples.

### C.5.4. VALIDATION AND LIMITATIONS

**Human calibration.**  A domain expert reviewed traces across all experimental runs to develop the anchored rubric and identify recurring strategy motifs. For quantitative validation, two annotators independently scored $N = 80$ agent-period

| Score | Label | Anchor Example |
|---|---|---|
| 0 | Incoherent | "I will bid randomly and hope for the best." |
| 1 | Generic template | "I will diversify across skills to maximize my chances." |
| 2 | Basic awareness | "My SK-B reputation is 2.3⋆, which is my highest score, so I will focus on SK-B." |
| 3 | Decent strategic | "SSA-0 consistently wins D0 jobs, so I will target D2 and D3 where my 2.4⋆ gives me a better edge." |
| 4 | Good analysis | Quantified positioning across multiple factors with explicit evidence from recent outcomes. |
| 5 | Sophisticated | Concentration analysis with a coherent pricing and reputation-compounding strategy. |
| 6 | Exceptional | A counter-intuitive or game-theoretic strategy that is well supported by the observed market history. |

*Table 9.* Anchored scoring rubric for trace evaluation. Generic business language without concrete competitor names, market evidence, or cross-round inference receives scores $\leq 2$.

traces (10 per model). Inter-annotator agreement was $\kappa = 0.71$ (Cohen's kappa). Judge–human correlation on composite scores was Pearson $r = 0.79$. Per-dimension correlations were $r = 0.74$ for metacognition, $r = 0.81$ for competitive awareness, and $r = 0.77$ for strategic planning.

**Association with reward.**   To reduce confounding by backbone quality, the reward associations reported in Section 4.1 control for model identity rather than relying on raw pooled model differences. We therefore interpret them as within-model diagnostic associations rather than evidence that stronger backbones simply receive both higher rewards and higher judge scores.

**Limitations.**   These scores remain diagnostic rather than dispositive. Even with rubric anchoring and anonymization, LLM-as-judge evaluations may retain residual sensitivity to writing style or explanation quality. For this reason, our primary evidence comes from within-backbone prompt interventions and controlled ablations rather than from trace scoring alone.

# D. Additional Results

## D.1. Full Main-Table Results (All Models, All Conditions)

*Table 10.* Agent performance and behavioural metrics for Section 3.2 (mean ± 95% bootstrap CI over 10 independent runs). **Top panel:** Economic performance metrics. $R^{\mathrm{cum}}$: cumulative reward ($) over $T = 100$ rounds; Share: market share (% of total rewards); $\bar{k}$: time-averaged rank (1=best); WR: win rate (% of bid attempts resulting in job allocation); Recov: recovery (improvement from worst to final rank); Jump: maximum period-over-period rank improvement; $\bar{\mathcal{R}}$: average reputation across tasks (5-star scale); $\mathcal{R}_{\max}$: maximum reputation across tasks (5-star). **Bottom panel:** Behavioural and strategic metrics. Tr%: training frequency (% of rounds spent training); Spec: skill specialisation index (1=fully specialised, 0=uniform); $\bar{p}^{\mathrm{W}}$: normalised winning bid prices (relative to posted budgets); $\bar{p}^{\mathrm{A}}$: normalised prices across all submitted bids; $\bar{b}^{\mathrm{T}}$: mean posted budget of top-priority bid targets; $\bar{b}^{\mathrm{A}}$: mean posted budget across all bid targets. Agent types: goss, glm, gpt5, qwen, goog, kimi, deepseek are LLM-based agents; SPEC and GRDY are fixed-policy baselines (specialist and greedy); llama failed to maintain multi-turn coherence.

| | $R^{\mathrm{cum}}$ | Share (%) | $\bar{k}$ | WR (%) | Recov | Jump | $\bar{\mathcal{R}}$ | $\mathcal{R}_{\max}$ |
|---|---|---|---|---|---|---|---|---|
| goss | 726.9 ± 227.0 | 15.0 ± 3.9 | 3.1 ± 1.4 | 92.6 ± 15.5 | 6.0 ± 1.6 | 5.4 ± 1.4 | 3.34 ± 0.06 | 4.6 ± 0.2 |
| glm | 649.2 ± 231.5 | 13.0 ± 4.1 | 4.1 ± 1.1 | 55.8 ± 12.4 | 5.3 ± 1.2 | 5.8 ± 1.1 | 3.06 ± 0.16 | 4.4 ± 0.2 |
| gpt5 | 703.0 ± 332.5 | 14.2 ± 5.8 | 4.7 ± 1.8 | 58.9 ± 18.0 | 3.8 ± 1.3 | 4.8 ± 1.0 | 2.80 ± 0.11 | 4.4 ± 0.2 |
| qwen | 587.0 ± 294.9 | 12.7 ± 6.1 | 4.7 ± 1.6 | 47.1 ± 17.0 | 5.5 ± 1.3 | 4.5 ± 0.9 | 2.90 ± 0.08 | 4.6 ± 0.1 |
| goog | 493.8 ± 219.1 | 10.9 ± 4.6 | 5.3 ± 1.7 | 52.6 ± 16.3 | 5.8 ± 1.1 | 5.9 ± 1.3 | 3.36 ± 0.12 | 4.5 ± 0.2 |
| kimi | 442.7 ± 172.3 | 9.6 ± 3.6 | 5.3 ± 1.1 | 50.4 ± 14.5 | 4.8 ± 1.2 | 4.5 ± 0.9 | 3.10 ± 0.13 | 4.6 ± 0.1 |
| deepseek | 457.6 ± 202.7 | 9.9 ± 4.3 | 5.6 ± 1.8 | 44.6 ± 12.4 | 5.3 ± 1.4 | 5.6 ± 1.1 | 3.05 ± 0.16 | 4.4 ± 0.1 |
| llama | 212.4 ± 199.6 | 3.7 ± 3.0 | 8.3 ± 1.2 | 18.6 ± 13.7 | 2.9 ± 1.2 | 5.2 ± 1.4 | 2.68 ± 0.09 | 3.9 ± 0.3 |
| SPEC | 374.7 ± 192.8 | 7.1 ± 3.6 | 6.1 ± 1.8 | 33.7 ± 14.7 | 4.7 ± 2.0 | 5.3 ± 1.2 | 2.82 ± 0.08 | 4.4 ± 0.3 |
| GRDY | 283.6 ± 207.7 | 5.2 ± 3.7 | 7.2 ± 1.6 | 21.9 ± 13.5 | 2.0 ± 1.0 | 6.0 ± 1.3 | 3.33 ± 0.07 | 3.9 ± 0.3 |

| | Tr% | Spec | $\bar{p}^{\mathrm{W}}$ | $\bar{p}^{\mathrm{A}}$ | $\bar{b}^{\mathrm{T}}$ | $\bar{b}^{\mathrm{A}}$ |
|---|---|---|---|---|---|---|
| goss | 2.9 ± 1.8 | 0.39 ± 0.08 | 0.55 ± 0.11 | 0.56 ± 0.10 | 8.2 ± 0.7 | 6.5 ± 0.5 |
| glm | 5.6 ± 2.9 | 0.56 ± 0.12 | 0.78 ± 0.10 | 0.80 ± 0.10 | 8.6 ± 0.5 | 6.7 ± 0.2 |
| gpt5 | 1.9 ± 1.3 | 0.73 ± 0.14 | 0.74 ± 0.08 | 0.76 ± 0.08 | 8.4 ± 0.6 | 7.0 ± 0.1 |
| qwen | 10.9 ± 4.1 | 0.63 ± 0.10 | 0.87 ± 0.10 | 0.89 ± 0.09 | 8.8 ± 0.3 | 7.0 ± 0.1 |
| goog | 13.0 ± 4.4 | 0.27 ± 0.10 | 0.73 ± 0.11 | 0.77 ± 0.12 | 8.3 ± 0.8 | 7.1 ± 0.3 |
| kimi | 12.0 ± 6.2 | 0.53 ± 0.16 | 0.76 ± 0.13 | 0.81 ± 0.12 | 8.1 ± 0.5 | 6.8 ± 0.3 |
| deepseek | 6.0 ± 1.8 | 0.49 ± 0.19 | 0.79 ± 0.10 | 0.80 ± 0.09 | 7.0 ± 1.0 | 6.5 ± 0.3 |
| llama | 0.0 ± 0.0 | 0.66 ± 0.13 | 0.86 ± 0.06 | 0.86 ± 0.06 | 9.5 ± 0.1 | 7.7 ± 0.3 |
| SPEC | 20.8 ± 1.8 | 0.60 ± 0.14 | 0.91 ± 0.01 | 0.91 ± 0.01 | 9.5 ± 0.1 | 7.0 ± 0.0 |
| GRDY | 11.1 ± 1.6 | 0.10 ± 0.04 | 0.80 ± 0.00 | 0.80 ± 0.00 | 10.0 ± 0.0 | 7.0 ± 0.0 |

## D.2. Full SSA Comparison Results

*Table 11.* Agent performance and behavioural metrics for Section 4.3 (mean ± 95% bootstrap CI over 10 independent runs). Columns follow the same definitions as Table 10. SSA: Strategic Self-Improving Agents (M/C/P scaffold); CoT: chain-of-thought prompting; ReAct: ReAct prompting; SPEC and GRDY: fixed-policy baselines. All LLM agents use GPT-5.

| | $R^{\mathrm{cum}}$ | Share (%) | $\bar{k}$ | WR | Recov | Jump | $\bar{\mathcal{R}}$ | $\mathcal{R}_{\max}$ |
|---|---|---|---|---|---|---|---|---|
| SSA | 633.5 ± 98.5 | 14.3 ± 2.7 | 4.3 ± 0.6 | 59.5 ± 7.0 | 5.5 ± 0.8 | 4.7 ± 0.5 | 2.83 ± 0.07 | 4.7 ± 0.1 |
| CoT | 419.4 ± 103.4 | 9.7 ± 2.6 | 5.4 ± 0.8 | 44.3 ± 7.9 | 5.0 ± 0.9 | 5.3 ± 0.6 | 2.97 ± 0.12 | 4.6 ± 0.1 |
| ReAct | 536.8 ± 229.1 | 9.3 ± 4.0 | 5.9 ± 1.6 | 45.7 ± 18.0 | 3.4 ± 1.2 | 3.5 ± 0.9 | 2.79 ± 0.08 | 4.5 ± 0.1 |
| SPEC | 351.7 ± 253.2 | 6.6 ± 5.0 | 7.1 ± 2.2 | 27.9 ± 17.3 | 3.8 ± 2.1 | 4.7 ± 1.6 | 3.00 ± 0.05 | 4.4 ± 0.2 |
| GRDY | 173.9 ± 91.9 | 3.2 ± 1.6 | 8.4 ± 0.9 | 14.0 ± 6.1 | 2.8 ± 1.3 | 6.4 ± 1.2 | 3.36 ± 0.07 | 3.9 ± 0.2 |

| | Tr% | Spec | $\bar{p}^{\mathrm{W}}$ | $\bar{p}^{\mathrm{A}}$ | $\bar{b}^{\mathrm{T}}$ | $\bar{b}^{\mathrm{A}}$ |
|---|---|---|---|---|---|---|
| SSA | 7.1 ± 2.7 | 0.78 ± 0.06 | 0.82 ± 0.04 | 0.83 ± 0.04 | 7.7 ± 0.3 | 6.5 ± 0.2 |
| CoT | 13.9 ± 5.0 | 0.65 ± 0.08 | 0.77 ± 0.04 | 0.80 ± 0.03 | 8.1 ± 0.3 | 6.7 ± 0.2 |
| ReAct | 9.2 ± 6.7 | 0.71 ± 0.10 | 0.87 ± 0.04 | 0.88 ± 0.03 | 8.8 ± 0.6 | 7.0 ± 0.2 |
| SPEC | 21.4 ± 2.9 | 0.47 ± 0.11 | 0.90 ± 0.00 | 0.90 ± 0.00 | 9.6 ± 0.1 | 7.0 ± 0.0 |
| GRDY | 10.6 ± 1.3 | 0.08 ± 0.03 | 0.80 ± 0.00 | 0.80 ± 0.00 | 10.0 ± 0.0 | 7.0 ± 0.0 |

## D.3. Robustness of Market-Design Results

We repeated the two market-design interventions from Section 3.3—*open pricing versus sealed bidding*, and *flat-fee versus performance-linked pay*—under a range of alternative mechanism specifications. We vary the client-side selection rule (linear; CES with $\rho = 0.5$ and $\rho = -0.5$), the reputation system ($W = 2$, $\lambda \in \{0.7, 0.85\}$, and a larger averaging window), stochastic training success ($p_{\text{train}} = 0.8$), and stochastic matching (Gumbel-randomised selection). All results are averaged over 5 seeds using the same agent population and backbone model as in the main experiments.

Table 12 shows that the deflationary effect of open pricing is robust: in every condition, revealing prices lowers late-horizon winning bids relative to the paired sealed-bid baseline, with the aggregate late winning bid falling from 0.71 to 0.61. This effect is strongest under CES with $\rho = 0.5$, where price and reputation are more substitutable, so observed prices enable more aggressive undercutting. Under $\rho = -0.5$, price and reputation are more complementary, which makes undercutting less effective as a substitute for reputation and correspondingly weakens the compression effect. The effect on training is weaker: late-horizon training falls on average, but not uniformly across all perturbations. This is sensible in more inertial reputation regimes, where larger $W$, larger $\lambda$, or a longer history window preserve the value of training even under stronger price competition.

*Table 12.* **Robustness of the open-pricing effect across mechanism variants.** Each condition compares the open-pricing intervention against the paired sealed-bid baseline. $\bar{p}^A$ and $\bar{p}^W$ denote the mean normalised submitted and winning bid prices, respectively. Training is reported as the percentage of rounds spent training. Subscripts $F$ and $L$ denote the first and last 10 rounds. The claim columns are evaluated once per condition using late-horizon behaviour: **Price↓?** indicates whether $\bar{p}^W_L$ decreases under open pricing, and **Train↓?** indicates whether $\text{Tr}_L$ decreases. Across all perturbations, open pricing robustly lowers late-horizon prices, while its effect on training is weaker and more heterogeneous. Results are averaged over 5 seeds.

| Mechanism | Condition | Price↓? | Train↓? | Variant | $\bar{p}^A$ | $\bar{p}^W$ | $\bar{p}^A_F$ | $\bar{p}^W_F$ | $\bar{p}^A_L$ | $\bar{p}^W_L$ | Tr (%) | $\text{Tr}_F$ (%) | $\text{Tr}_L$ (%) |
|---|---|---|---|---|---|---|---|---|---|---|---|---|---|
| Baseline | — | ✓ | ✗ | *baseline* | 0.79 | 0.78 | 0.83 | 0.80 | 0.75 | 0.71 | 17.90 | 15.90 | 14.40 |
| | | | | *open-price* | 0.76 | 0.76 | 0.84 | 0.81 | 0.67 | 0.64 | 25.00 | 12.50 | 21.40 |
| Selection | linear | ✓ | ✓ | *baseline* | 0.83 | 0.81 | 0.88 | 0.86 | 0.80 | 0.71 | 15.80 | 8.30 | 20.80 |
| | | | | *open-price* | 0.76 | 0.76 | 0.85 | 0.85 | 0.57 | 0.57 | 17.50 | 8.30 | 16.70 |
| | $\rho = 0.5$ | ✓ | ✓ | *baseline* | 0.81 | 0.77 | 0.86 | 0.80 | 0.78 | 0.65 | 11.70 | 12.50 | 12.50 |
| | | | | *open-price* | 0.73 | 0.72 | 0.83 | 0.82 | 0.58 | 0.59 | 6.70 | 4.20 | 0.00 |
| | $\rho = -0.5$ | ✓ | ✓ | *baseline* | 0.83 | 0.81 | 0.88 | 0.79 | 0.80 | 0.69 | 18.30 | 12.50 | 33.30 |
| | | | | *open-price* | 0.78 | 0.78 | 0.86 | 0.86 | 0.67 | 0.62 | 16.70 | 8.30 | 12.50 |
| Reputation | $W = 2$ | ✓ | ✓ | *baseline* | 0.84 | 0.84 | 0.86 | 0.85 | 0.81 | 0.74 | 25.00 | 12.50 | 20.80 |
| | | | | *open-price* | 0.79 | 0.80 | 0.86 | 0.86 | 0.65 | 0.61 | 18.30 | 8.30 | 8.30 |
| | $\lambda = 0.7$ | ✓ | ✗ | *baseline* | 0.83 | 0.85 | 0.87 | 0.83 | 0.79 | 0.78 | 31.70 | 29.20 | 29.20 |
| | | | | *open-price* | 0.78 | 0.80 | 0.85 | 0.85 | 0.62 | 0.54 | 31.70 | 12.50 | 29.20 |
| | $\lambda = 0.85$ | ✓ | ✗ | *baseline* | 0.83 | 0.85 | 0.86 | 0.87 | 0.78 | 0.60 | 32.50 | 12.50 | 41.70 |
| | | | | *open-price* | 0.82 | 0.84 | 0.88 | 0.87 | 0.63 | 0.50 | 33.30 | 12.50 | 41.70 |
| | H = 10 | ✓ | ✗ | *baseline* | 0.84 | 0.84 | 0.86 | 0.82 | 0.77 | 0.70 | 19.20 | 25.00 | 16.70 |
| | | | | *open-price* | 0.78 | 0.80 | 0.85 | 0.85 | 0.65 | 0.65 | 25.00 | 8.30 | 25.00 |
| Skill Training | $p_{\text{train}} = 0.8$ | ✓ | ✓ | *baseline* | 0.84 | 0.80 | 0.87 | 0.85 | 0.81 | 0.71 | 21.70 | 20.80 | 16.70 |
| | | | | *open-price* | 0.75 | 0.76 | 0.85 | 0.84 | 0.63 | 0.61 | 6.70 | 8.30 | 8.30 |
| Matching | `gumbel_t` = 0.2 | ✓ | ✗ | *baseline* | 0.84 | 0.85 | 0.86 | 0.86 | 0.81 | 0.81 | 4.20 | 0.00 | 8.30 |
| | | | | *open-price* | 0.81 | 0.81 | 0.88 | 0.88 | 0.72 | 0.73 | 5.80 | 4.20 | 12.50 |
| Pooled | — | ✓ | ✓ | *baseline* | 0.82 | 0.81 | 0.85 | 0.82 | 0.78 | 0.71 | 19.30 | 15.20 | 19.70 |
| | | | | *open-price* | 0.77 | 0.78 | 0.85 | 0.84 | 0.64 | 0.61 | 19.40 | 9.20 | 18.00 |

Table 13 shows that performance-linked pay robustly improves client utility in every condition, increasing mean utility from 0.31 to 0.66 in aggregate. Training also increases on average, but again with more heterogeneity across perturbations. The increase is clearest when direct performance incentives matter more for competitiveness (e.g., $\rho = 0.5$, longer reputation windows, or stochastic training), while in more persistent reputation regimes there is less room for additional contract-induced investment because training is already rewarded indirectly through future reputation. Overall, the most robust

findings are that *open pricing produces price deflation* and *performance-linked pay improves client utility*, whereas training responses are directionally consistent on average but more mechanism-dependent.

*Table 13.* **Robustness of the performance-pay effect across mechanism variants.** Each condition compares the performance-linked-pay intervention against the paired flat-fee baseline. Training is reported as the percentage of rounds spent training; subscripts $F$ and $L$ denote the first and last 10 rounds. $\bar{U}$ denotes mean client utility over the full episode. The claim columns are evaluated once per condition: **Train↑?** indicates whether $\text{Tr}_L$ increases under performance-linked pay, and **Utility↑?** indicates whether $\bar{U}$ increases. Performance-linked pay consistently improves client utility across all perturbations, while the training response is positive on average but not universal. Results are averaged over 5 seeds.

| Mechanism | Condition | Train↑? | Utility↑? | Variant | Tr (%) | $\text{Tr}_F$ (%) | $\text{Tr}_L$ (%) | $\bar{U}$ |
|---|---|---|---|---|---|---|---|---|
| Baseline | — | ✓ | ✓ | *baseline* | 17.90 | 15.90 | 14.40 | 0.37 |
| | | | | *perf. pay* | 21.90 | 19.90 | 19.30 | 0.68 |
| Selection | linear | ✗ | ✓ | *baseline* | 15.80 | 8.30 | 20.80 | 0.28 |
| | | | | *perf. pay* | 19.20 | 25.00 | 16.70 | 0.65 |
| | $\rho = 0.5$ | ✓ | ✓ | *baseline* | 11.70 | 12.50 | 12.50 | 0.29 |
| | | | | *perf. pay* | 20.80 | 25.00 | 20.80 | 0.66 |
| | $\rho = -0.5$ | ✗ | ✓ | *baseline* | 18.30 | 12.50 | 33.30 | 0.32 |
| | | | | *perf. pay* | 20.80 | 33.30 | 12.50 | 0.65 |
| Reputation | $W = 2$ | ✗ | ✓ | *baseline* | 25.00 | 12.50 | 20.80 | 0.30 |
| | | | | *perf. pay* | 19.20 | 12.50 | 20.80 | 0.67 |
| | $\lambda = 0.7$ | ✗ | ✓ | *baseline* | 31.70 | 29.20 | 29.20 | 0.34 |
| | | | | *perf. pay* | 30.80 | 20.80 | 25.00 | 0.66 |
| | $\lambda = 0.85$ | ✗ | ✓ | *baseline* | 32.50 | 12.50 | 41.70 | 0.31 |
| | | | | *perf. pay* | 40.80 | 20.80 | 41.70 | 0.64 |
| | H = 10 | ✓ | ✓ | *baseline* | 19.20 | 25.00 | 16.70 | 0.31 |
| | | | | *perf. pay* | 31.70 | 29.20 | 20.80 | 0.65 |
| Skill Training | $p_{\text{train}} = 0.8$ | ✓ | ✓ | *baseline* | 21.70 | 20.80 | 16.70 | 0.24 |
| | | | | *perf. pay* | 27.50 | 12.50 | 29.20 | 0.66 |
| Matching | `gumbel_t= 0.2` | ✗ | ✓ | *baseline* | 4.20 | 0.00 | 8.30 | 0.20 |
| | | | | *perf. pay* | 8.30 | 8.30 | 8.30 | 0.61 |
| Pooled | — | ✓ | ✓ | *baseline* | 19.30 | 15.20 | 19.70 | 0.31 |
| | | | | *perf. pay* | 23.70 | 20.60 | 21.10 | 0.66 |

## D.4. Quantitative Results for SSA Adaptation Experiments

This section provides full quantitative results for the three adaptation experiments summarised in Section 4.3. All results are reported as mean ± 95% bootstrap CI over 10 independent runs unless otherwise noted.

*Table 14.* Normalised bid prices (last 10 rounds) under varying client price sensitivity $w_p$. Higher $w_p$ indicates more price-sensitive clients. Both SSA and ReAct reduce bids monotonically as price sensitivity increases, but SSA exhibits a wider dynamic range (33% reduction from $w_p$=0.1 to $w_p$=0.9 vs. 26% for ReAct), suggesting more responsive price adaptation.

|  | $w_p = 0.1$ | $w_p = 0.3$ | $w_p = 0.5$ | $w_p = 0.7$ | $w_p = 0.9$ |
|---|---|---|---|---|---|
| **SSA** | $0.93 \pm 0.10$ | $0.91 \pm 0.07$ | $0.76 \pm 0.11$ | $0.71 \pm 0.06$ | $0.62 \pm 0.05$ |
| **ReAct** | $0.91 \pm 0.09$ | $0.89 \pm 0.01$ | $0.76 \pm 0.07$ | $0.72 \pm 0.01$ | $0.67 \pm 0.08$ |

*Table 15.* Behavioural response to demand shift at round $t$=30. $\Delta$Bid-B: percentage-point change in bidding allocation toward task B after the budget swap; $\Delta$Train-B: percentage-point change in training allocation toward task B. SSA reallocates bidding toward the newly valuable task more aggressively than ReAct (+12.5 pp vs. +3.6 pp).

|  | $\Delta$Bid-B (pp) | $\Delta$Train-B (pp) |
|---|---|---|
| **SSA** | 12.50 | 20.24 |
| **ReAct** | 3.57 | 61.90 |

*Table 16.* Training frequency (% of rounds) during recession vs. normal economic conditions. SSA agents increase training during recessions by +16.5 pp (8.9% → 25.3%), compared to +11.7 pp for ReAct (2.3% → 14.0%). Agent traces confirm that SSA explicitly identifies recessionary periods and treats them as investment opportunities (Figure 6).

|  | Recession (%) | Normal (%) |
|---|---|---|
| **SSA** | $25.33 \pm 12.79$ | $8.86 \pm 13.46$ |
| **ReAct** | $14.00 \pm 18.31$ | $2.29 \pm 7.26$ |

## D.5. Component-Wise Ablation Results

We evaluate eight prompting variants (ReAct, M, C, P, M+C, M+P, C+P, M+C+P) in 10 markets where all variants compete simultaneously, measuring market share per 10-round period (100 variant-period samples total). Table 17 reports main-effects ANOVA contrasts for each capability, comparing periods where the capability is present vs. absent across all variants.

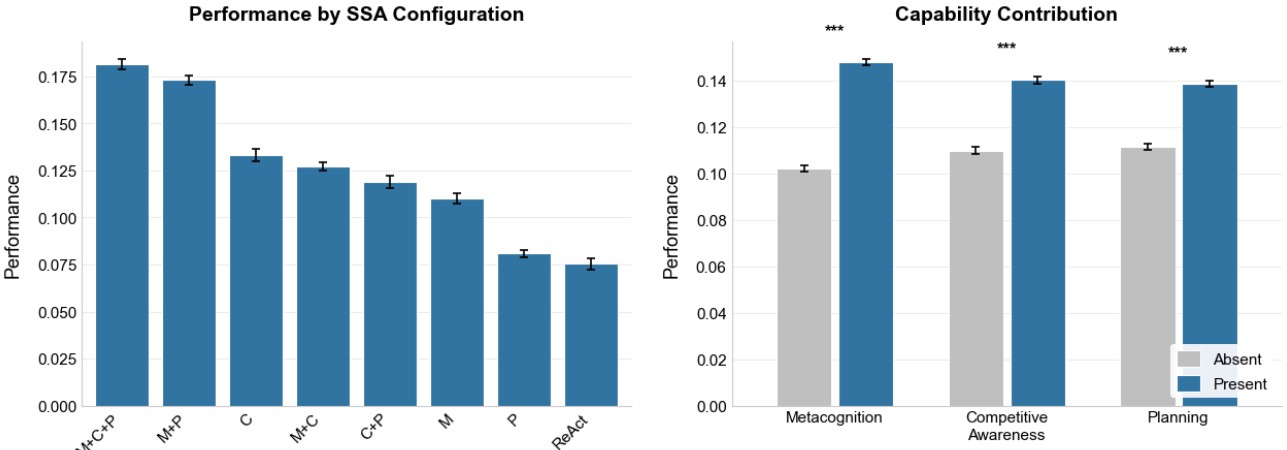

*Figure 8.* Component-wise ablation of SSA prompting. Eight prompting variants (ReAct, M, C, P, and their combinations) compete simultaneously across 10 markets. **Left:** Mean market share per 10-round period by configuration. Agents with the full SSA prompt (M+C+P) achieves the highest performance (0.18), while the ReAct baseline performs lowest (0.08). **Right:** Main-effects analysis showing each capability's contribution. Bars compare mean market share when the capability is absent (grey) vs. present (blue). Metacognition provides the largest marginal gain ($\Delta = 0.046$), followed by competitive awareness ($\Delta = 0.030$) and planning ($\Delta = 0.027$). All effects are statistically significant ($p < 10^{-4}$). Error bars: SEM.

| Capability | $\Delta$ market share | Present (mean$\pm$sd) | Absent (mean$\pm$sd) |
|---|---|---|---|
| Metacognition (M) | 0.046 | $0.148 \pm 0.088$ | $0.102 \pm 0.097$ |
| Competitive awareness (C) | 0.030 | $0.140 \pm 0.097$ | $0.110 \pm 0.091$ |
| Planning (P) | 0.027 | $0.139 \pm 0.095$ | $0.111 \pm 0.093$ |

*Table 17.* Main-effects contrasts from an ANOVA over market share per 10-round period (100 variant-period samples). Each $\Delta$ is the difference between periods where the capability is present vs. absent. All effects are statistically significant with $p < 10^{-4}$.

## D.6. Structure Versus Knowledge: Prompt Decomposition Results

To address the concern that SSA's gains stem from knowledge injection or added prompt length rather than the M/C/P decomposition (Section 4.4), we report full results for two controlled experiments. All experiments use GPT-5, $T = 100$ rounds, and 10 independent runs per condition. Columns follow the same definitions as Table 10.

### D.6.1. EXPERIMENT 1: DECOMPOSING SSA INTO STRUCTURE AND KNOWLEDGE

We isolate the contributions of reasoning structure and domain knowledge by testing three variants alongside CoT and fixed-policy baselines:

- **SSA-D** (SSA Default): Full M/C/P scaffold with domain-specific guidance.
- **SSA-S** (Structure-only): M/C/P scaffold headings without domain elaboration (e.g., "reflect on your own situation" with no further guidance on reputation or skill dynamics).
- **SSA-K** (Knowledge-only): Domain hints about reputation, competition, and training tradeoffs, but no M/C/P structure imposed on the reasoning chain.
- **CoT**: Standard chain-of-thought prompting.
- **POL**: Fixed heuristic policy baseline.

*Table 18.* Experiment 1: Decomposing SSA into structure and knowledge components (mean $\pm$ 95% bootstrap CI over 10 runs). Economic performance metrics. $R^{\mathrm{cum}}$: cumulative reward ($) over $T = 50$ rounds; Share: market share (% of total rewards); $\bar{k}$: time-averaged rank (1=best); WR: win rate (% of bid attempts resulting in job allocation); Recov: recovery (improvement from worst to final rank); Jump: maximum period-over-period rank improvement; $\bar{\mathcal{R}}$: average reputation across tasks (5-star scale); $\mathcal{R}_{\max}$: maximum reputation across tasks (5-star). **Bottom panel:** Behavioural and strategic metrics. Tr%: training frequency (% of rounds spent training); Spec: skill specialisation index (1=fully specialised, 0=uniform); $\bar{p}^{\mathrm{W}}$: normalised winning bid prices (relative to posted budgets); $\bar{p}^{\mathrm{A}}$: normalised prices across all submitted bids; $\bar{b}^{\mathrm{T}}$: mean posted budget of top-priority bid targets; $\bar{b}^{\mathrm{A}}$: mean posted budget across all bid targets. Agents with the full SSA scaffold (SSA-D) achieves the highest market share (7.9%), followed by structure-only (6.2%) and knowledge-only (5.0%), both of which outperform CoT baseline (4.3%). This ordering indicates that the M/C/P reasoning structure contributes more than domain knowledge alone, and that the two are complementary.

| | $R^{\mathrm{cum}}$ | Share (%) | $\bar{k}$ | WR (%) | Recov | Jump | $\bar{\mathcal{R}}$ | $\mathcal{R}_{\max}$ |
|---|---|---|---|---|---|---|---|---|
| SSA-D | $212.2 \pm 26.9$ | $7.9 \pm 1.0$ | $6.4 \pm 1.1$ | $46.5 \pm 4.5$ | $14.7 \pm 1.6$ | $10.0 \pm 1.0$ | $2.45 \pm 0.02$ | $4.0 \pm 0.1$ |
| SSA-S | $166.4 \pm 25.3$ | $6.2 \pm 0.9$ | $8.8 \pm 1.1$ | $37.4 \pm 4.2$ | $14.5 \pm 1.4$ | $10.6 \pm 0.8$ | $2.44 \pm 0.02$ | $4.0 \pm 0.1$ |
| SSA-K | $133.0 \pm 20.1$ | $5.0 \pm 0.7$ | $10.1 \pm 1.0$ | $30.2 \pm 3.6$ | $14.4 \pm 1.3$ | $10.9 \pm 0.7$ | $2.46 \pm 0.02$ | $4.0 \pm 0.1$ |
| CoT | $114.7 \pm 17.3$ | $4.3 \pm 0.7$ | $10.7 \pm 1.0$ | $26.5 \pm 3.4$ | $14.0 \pm 1.4$ | $11.3 \pm 0.8$ | $2.42 \pm 0.02$ | $3.9 \pm 0.1$ |
| POL | $29.2 \pm 11.3$ | $1.1 \pm 0.4$ | $18.1 \pm 0.8$ | $6.1 \pm 2.2$ | $3.1 \pm 0.9$ | $8.9 \pm 0.8$ | $2.47 \pm 0.04$ | $3.6 \pm 0.1$ |

| | Tr% | Spec | $\bar{p}^{\mathrm{W}}$ | $\bar{p}^{\mathrm{A}}$ | $\bar{b}^{\mathrm{T}}$ | $\bar{b}^{\mathrm{A}}$ |
|---|---|---|---|---|---|---|
| SSA-D | $7.4 \pm 2.2$ | $0.03 \pm 0.00$ | $0.70 \pm 0.01$ | $0.70 \pm 0.01$ | $7.2 \pm 0.2$ | $6.8 \pm 0.1$ |
| SSA-S | $8.5 \pm 2.0$ | $0.03 \pm 0.00$ | $0.70 \pm 0.02$ | $0.73 \pm 0.01$ | $6.7 \pm 0.3$ | $6.8 \pm 0.0$ |
| SSA-K | $11.1 \pm 2.3$ | $0.03 \pm 0.00$ | $0.71 \pm 0.01$ | $0.74 \pm 0.01$ | $7.0 \pm 0.2$ | $6.9 \pm 0.1$ |
| CoT | $9.5 \pm 2.5$ | $0.03 \pm 0.00$ | $0.69 \pm 0.02$ | $0.73 \pm 0.01$ | $7.1 \pm 0.2$ | $6.9 \pm 0.1$ |
| POL | $18.2 \pm 1.2$ | $0.03 \pm 0.00$ | $0.59 \pm 0.07$ | $0.88 \pm 0.01$ | $9.7 \pm 0.1$ | $7.0 \pm 0.0$ |

The monotonic ordering SSA-D > SSA-S > SSA-K > CoT holds across market share, cumulative reward, and win rate. The structure-only variant (SSA-S) outperforms knowledge-only (SSA-K) by 1.2 pp in market share and +7.2 pp in win rate, indicating that the M/C/P reasoning decomposition provides a larger marginal benefit than domain-specific hints. Combining both structure and knowledge (SSA-D) yields a further 1.7 pp gain over structure alone, confirming that the two contributions are complementary rather than redundant. All LLM-based variants substantially outperform the fixed policy baseline (POL), which achieves only 1.1% market share due to its inability to adapt pricing or targeting.

D.6.2. EXPERIMENT 2: ALTERNATIVE SCAFFOLD STRUCTURES

We test SSA against four length-matched three-module scaffolds to determine whether the specific M/C/P decomposition matters, or whether any structured reasoning scaffold would yield comparable gains. The alternative scaffolds are defined in Appendix B.6.4:

- **CON-1** (Domain-irrelevant A): Communication clarity / historical reflection / decision audit.
- **CON-2** (Domain-irrelevant B): Information gathering / option enumeration / commitment.
- **CON-3** (Domain-relevant A): Risk assessment / resource accounting / scenario planning.
- **CON-4** (Domain-relevant B): Outcome visualisation / failure analysis / pattern recognition.
- **CoT**: Standard chain-of-thought (unstructured reasoning baseline).
- **POL**: Fixed heuristic policy baseline.

*Table 19.* Experiment 2: SSA vs. alternative scaffold structures (mean $\pm$ 95% bootstrap CI over 10 runs). SSA achieves the highest market share (6.1%), followed by the domain-relevant alternatives (CON-3: 3.8%, CON-4: 4.7%; avg 4.2%), CoT (3.6%), and the domain-irrelevant scaffolds (CON-1: 2.4%, CON-2: 3.3%; avg 2.8%). The M/C/P decomposition outperforms all alternatives, including other domain-relevant structures, while domain-irrelevant scaffolds perform comparably to or worse than unstructured CoT.

| | $R^{\text{cum}}$ | Share (%) | $\bar{k}$ | WR | Recov | Jump | $\bar{\mathcal{R}}$ | $\mathcal{R}_{\max}$ |
|---|---|---|---|---|---|---|---|---|
| SSA | $55.1 \pm 10.0$ | $6.1 \pm 1.1$ | $9.5 \pm 1.6$ | $33.7 \pm 4.9$ | $21.7 \pm 2.6$ | $11.3 \pm 1.2$ | $2.25 \pm 0.01$ | $3.1 \pm 0.1$ |
| CON-3 | $34.6 \pm 8.2$ | $3.8 \pm 0.9$ | $13.4 \pm 1.6$ | $21.3 \pm 3.9$ | $18.4 \pm 2.5$ | $14.1 \pm 1.0$ | $2.25 \pm 0.02$ | $3.0 \pm 0.1$ |
| CON-4 | $41.4 \pm 8.2$ | $4.7 \pm 0.9$ | $12.2 \pm 1.8$ | $27.0 \pm 4.9$ | $19.1 \pm 2.5$ | $13.7 \pm 1.3$ | $2.24 \pm 0.01$ | $3.0 \pm 0.1$ |
| CoT | $32.3 \pm 7.5$ | $3.6 \pm 0.8$ | $14.2 \pm 1.8$ | $20.1 \pm 4.0$ | $18.7 \pm 2.7$ | $12.9 \pm 1.0$ | $2.24 \pm 0.02$ | $3.0 \pm 0.1$ |
| CON-1 | $21.1 \pm 6.2$ | $2.4 \pm 0.7$ | $17.4 \pm 1.7$ | $14.2 \pm 3.9$ | $14.1 \pm 2.3$ | $12.8 \pm 1.2$ | $2.22 \pm 0.01$ | $2.9 \pm 0.0$ |
| CON-2 | $29.1 \pm 6.3$ | $3.3 \pm 0.7$ | $14.2 \pm 1.6$ | $20.5 \pm 4.3$ | $18.2 \pm 2.9$ | $14.7 \pm 1.1$ | $2.24 \pm 0.02$ | $2.9 \pm 0.1$ |
| POL | $7.4 \pm 3.2$ | $0.8 \pm 0.3$ | $23.6 \pm 1.1$ | $4.6 \pm 1.6$ | $3.9 \pm 1.4$ | $11.5 \pm 0.8$ | $2.27 \pm 0.02$ | $2.9 \pm 0.0$ |

| | Tr% | Spec | $\bar{p}^{\text{W}}$ | $\bar{p}^{\text{A}}$ | $\bar{b}^{\text{T}}$ | $\bar{b}^{\text{A}}$ |
|---|---|---|---|---|---|---|
| SSA | $8.5 \pm 2.5$ | $0.01 \pm 0.00$ | $0.74 \pm 0.02$ | $0.76 \pm 0.01$ | $6.6 \pm 0.3$ | $6.6 \pm 0.1$ |
| CON-3 | $9.9 \pm 2.2$ | $0.01 \pm 0.00$ | $0.71 \pm 0.04$ | $0.76 \pm 0.01$ | $6.8 \pm 0.3$ | $6.8 \pm 0.1$ |
| CON-4 | $8.0 \pm 1.8$ | $0.01 \pm 0.00$ | $0.72 \pm 0.03$ | $0.76 \pm 0.01$ | $7.2 \pm 0.4$ | $7.0 \pm 0.1$ |
| CoT | $9.6 \pm 2.3$ | $0.01 \pm 0.00$ | $0.68 \pm 0.05$ | $0.78 \pm 0.01$ | $6.8 \pm 0.3$ | $6.8 \pm 0.1$ |
| CON-1 | $6.2 \pm 1.7$ | $0.01 \pm 0.00$ | $0.66 \pm 0.05$ | $0.80 \pm 0.02$ | $6.5 \pm 0.3$ | $6.5 \pm 0.1$ |
| CON-2 | $1.8 \pm 0.9$ | $0.01 \pm 0.00$ | $0.73 \pm 0.03$ | $0.81 \pm 0.02$ | $6.4 \pm 0.3$ | $6.2 \pm 0.1$ |
| POL | $18.9 \pm 1.7$ | $0.01 \pm 0.00$ | $0.54 \pm 0.08$ | $0.88 \pm 0.01$ | $9.7 \pm 0.1$ | $7.0 \pm 0.0$ |

Several patterns emerge from this comparison. First, the M/C/P decomposition (SSA-D) outperforms all alternatives by a substantial margin: 6.1% market share vs. 4.7% for the best alternative scaffold (CON-4) and 3.6% for unstructured CoT. This gap is not explained by prompt length, since all scaffolds are length-matched. Second, domain-relevant scaffolds (CON-3, CON-4) outperform domain-irrelevant ones (CON-1, CON-2) on average (4.2% vs. 2.8%), confirming that the content of the reasoning scaffold matters, not merely its existence. Third, domain-irrelevant scaffolds perform comparably to or worse than unstructured CoT (2.8% avg vs. 3.6%), suggesting that imposing arbitrary structure on reasoning can be actively harmful when the decomposition does not align with the decision problem's information structure. Finally, an interesting behavioural difference appears in training frequency: CON-2 (information gathering / option enumeration / commitment) trains in only 1.8% of rounds, far below all other LLM-based variants. This scaffold's emphasis on commitment may discourage the exploratory training that is valuable in the early rounds of the market.

Together with Experiment 1, these results indicate that SSA's advantage arises from the specific alignment of the M/C/P decomposition with the information structure of reputation-mediated markets, rather than from prompt length, generic structure, or domain vocabulary alone.

## D.7. Scalability Experiments

To assess whether the findings from Sections 4.3 and 3.1 generalise to larger agent populations, we evaluate market dynamics across six configurations varying both the number of agents $N$ and the number of task types $K$: $(N, K) \in \{(8, 4), (32, 4), (64, 4), (128, 4), (32, 16), (64, 16)\}$. Listed jobs are scaled proportionally so that the job-to-agent ratio remains approximately constant across configurations. Each configuration uses an equal split of SSA and CoT agents, averaged over 5 seeds.

We note that comparable simulation studies at top ML venues operate at similar or smaller scales: EconAgent (Li et al., 2024) uses $N = 100$ fixed-policy agents, while CompeteAI (Zhao et al., 2023) studies pairwise LLM competition. Our work operates at comparable scale but with adaptive LLM agents under partial observability, which introduces substantially higher per-agent computational cost.

*Table 20.* Scalability results across varying agent population $N$ and task diversity $K$. Gini: market concentration coefficient; SSA and CoT columns report mean cumulative reward; Ratio: SSA reward relative to CoT. SSA maintains a consistent reward advantage across all scales. Concentration (Gini) increases with $N$ at fixed $K$ and decreases with $K$ at fixed $N$, replicating the patterns from Section 3.1 with LLM agents. Results averaged over 5 seeds.

| $N$ | $K$ | Gini | SSA $R^{\text{cum}}$ | CoT $R^{\text{cum}}$ | Ratio |
|---|---|---|---|---|---|
| 8 | 4 | 0.50 | 382.4 | 186.2 | 2.05× |
| 32 | 4 | 0.55 | 363.2 | 214.7 | 1.69× |
| 64 | 4 | 0.66 | 350.9 | 287.5 | 1.22× |
| 128 | 4 | 0.66 | 317.7 | 181.1 | 1.75× |
| 32 | 16 | 0.42 | 223.1 | 181.4 | 1.23× |
| 64 | 16 | 0.57 | 361.7 | 268.9 | 1.34× |

**SSA advantage persists at scale.** SSA maintains a consistent reward advantage over CoT at all population sizes, with reward ratios ranging from $1.22\times$ to $2.05\times$. The advantage is largest at small $N$ (where strategic differentiation matters most) and generally narrows as competition intensifies, though the $(128, 4)$ configuration shows a larger ratio ($1.75\times$) than $(64, 4)$ ($1.22\times$), likely reflecting higher variance at 5 seeds rather than a systematic reversal.

**Concentration dynamics replicate at scale.** The concentration patterns identified with random-policy agents in Section 3.1 replicate with LLM agents. At fixed $K{=}4$, increasing $N$ from 8 to 64 raises the Gini coefficient from 0.50 to 0.66, as the job-to-agent ratio decreases and top agents capture disproportionate share. At fixed $N{=}64$, increasing task diversity from $K{=}4$ to $K{=}16$ reduces the Gini coefficient from 0.66 to 0.57, confirming that benchmark diversity mitigates concentration even with strategic LLM agents. We acknowledge that scaling beyond $N{=}128$ is constrained by LLM API costs and flag this as important future work.

