# OpenReview forum: "When AI Agents Compete for Jobs: Strategic Capabilities and Economic Dynamics of AI Labour Markets"
_ICML.cc/2026/Conference — ICML 2026 regular_

### Official Review · Reviewer_5M91 · 2026-02-17

**Soundness:** 2
**Presentation:** 1
**Significance:** 3
**Originality:** 3
**Overall Recommendation:** 2
**Confidence:** 3

**Summary:**

This paper constructed a formal framework as a testbed called "AI-work" to demonstrate AI labour market where multiple AI agents participate in a gig platform where they can bid for a job or train themselves. The platform will select bids according to pre-determiend rules. The results demonstrate that: (i) the framework is able to simulate market dynamics and most LLMs achieve higher rewards than vanilla policies;  (ii) the paper identifies 3 capabilities crucial for agents to make decisions in the labour market including Metacognition, Competitive awareness, and Strategic planning, and then use prompts to build SSA (Strategic Self-Improving Agents) to obtain better performance than regular CoT agents and ReAct Agents.

**Compliance With Llm Reviewing Policy:**

Affirmed.

**Final Justification:**

While the authors claim that a camera-ready version can solve all the problems on presentation and other concerns in weakness section, I still remain skeptical because it needs a significant revision, which seems to be more suitable to submit to a subsequent venue.

**Key Questions For Authors:**

See weaknesses

**Limitations:**

yes

**Strengths And Weaknesses:**

> Strengths

- The paper brings up an interesting and novel topic to simulate the AI labour market.
- The overall architecture of the framework and the performance metrics are clear.
- The 3 domains for strategic capabilities are reasonable.

> Weaknesses

- It is somewhat difficult to assess the paper based on the information in main sections because many mechanisms mentioned (e.g., the update formula for training, reputation, the details of building the SSA) are all in Appendix. And there are many "Appendix ??" in the main paper. I strongly encourage the authors to include all important and necessary justifications and quantitative formulas in the main paper. At least they should justify why the specified dynamics are reasonable and can reflect the real-world settings. Since the paper is highly relying on the appendix, I am not sure it is well-written enough.

- Meanwhile, the framework is a bit too artificial, where agents can only choose from two actions.

- The section of strategic capabilities seems to qualitative. What are the motivations for the authors to come up with the 3 domains in lines 301-315. Also, the correlation corresponds to the scores produced by LLM-as-judge, so it is somewhat meaningless.

Overall, I feel that the clarity of this paper should be improved

---

> ### Author Rebuttal · Authors · 2026-03-31
>
> We thank Reviewer 5M91 for the detailed review, and agree that the main paper's presentation was insufficient. The key equations and design rationale should have been in the main text, and the broken "Appendix ??" references made the manuscript more difficult to read. The intended appendix pointers are:
>
> Scoring/matching -> Appendix F.4 and G,
>
> Training/skill dynamics -> Appendix F.6,
>
> Reputation update -> Appendix F.7,
>
> Task implementations -> Appendix H,
>
> Prompts -> Appendix I.3,
>
> Experiment settings -> Appendix K.2,
>
> LLM-as-judge validation -> Appendix L.4,
>
> Full results -> Appendix N.
>
> We plan to substantially revise the presentation in future revisions, and move the core equations (Cobb-Douglas scoring, Beta-evidence reputation update, saturating skill update) into Section 2 with a Design Rationale subsection clarifying how AI-Work isolates core labor-market mechanisms. For this rebuttal period, we conducted additional experiments, with a one-page summary of new results: [image](https://ibb.co/RpsZHk5D). In response to your other comments:
>
> **1. The action space is binary and feels artificial.**
>
> While the meta-action is binary (BID vs. TRAIN), the effective decision space is substantially richer. Conditional on BID, an agent must jointly decide: (i) which subset of up to L=5 jobs to target from ~16 listings (C(16,5)=4368 subsets), (ii) a continuous bid price for each, and (iii) a strict preference ordering affecting allocation under stable matching, all under partial observability. TRAIN requires choosing which of K=4 skills to invest in. Each round thus involves four economic decisions: whether to bid or train, which jobs to target, how to price, and which skills to develop.
>
> Additionally, the BID vs. TRAIN structure itself is not artificial. We designed it to model the fundamental earn-now vs. invest-for-later tradeoff present on real platforms. On gig economy platforms such as Upwork and Fiverr, freelancers face similar choices with their time: to bid on available contracts (immediate income) or spend time upskilling / building portfolios / obtaining certifications (future competitiveness). The platform only observes bidding activity, off-platform investment is a hidden action, which is partly the structure we model. We will further discuss the action space and real-world relevance in future revisions of Section 2.
>
> **2. Motivation for the three capability domains.**
>
> We agree this needed stronger justification. The three domains correspond to three uncertainties induced by the market: uncertainty about one's own latent competitiveness (metacognition), about rivals/market state under hidden bids/skills (competitive awareness), and about how current actions shape future position (strategic planning). To reiterate our approach: (a) eight frontier LLMs reveal distinct strategic profiles; (b) M/C/P map to established ML concepts (calibration, opponent modeling, long-horizon planning); (c) the LLM-as-judge is diagnostic only, validated against human annotators (kappa=0.71, r=0.79; Appendix L.4). We agree that judge-based correlations alone would be weak evidence and do not use them as main support; (d) primary evidence comes from controlled SSA experiments (14.3% vs. 9.5% baseline market share) and component ablations (each of M, C, P significant at p<1e-4, Section 4.6).
>
> **3. Quantitative evidence on strategic capabilities.**
>
> Beyond the diagnostic trace analysis, we provide controlled quantitative evidence that SSA adapts more decisively than baselines across three market conditions ([figures](https://ibb.co/zVLCxVJc)):
>
> *Price sensitivity:* Sweeping client price weight w_p from 0.1 to 0.9, SSA's winning bids drop from 0.93 to 0.62, a wider dynamic range than baselines (0.91 to 0.67).
>
> *Demand shift:* At round t=30, we swap task budgets. SSA reallocates bidding toward the newly valuable task by +12.5pp vs. +3.6pp for ReAct. When outcompeted, SSA retreats to its prior specialization rather than persisting in low-return bidding.
>
> *Recession:* When budgets collapse to $1, SSA increases training to 25.3% of rounds vs. 8.9% normally, while ReAct increases from 2.3% to 14.0%. Agent traces confirm SSA explicitly treats recessions as investment periods and resumes aggressive bidding when budgets recover.
>
> Across all three conditions, SSA adapts in the economically expected direction and does so more decisively than baselines. These measured behavioral differences, combined with the component ablation and controlled prompting intervention, provide convergent evidence that the M/C/P capabilities are economically useful, not merely correlated with model quality via the LLM-as-judge pipeline.
>
> We hope these clarifications and experiments address the reviewer's concerns.

---

> > ### Author Rebuttal · Reviewer_5M91 · 2026-03-31
> >
> > Thanks for the rebuttal. I think the authors' rebuttal is beneficial in providing more information. However, just as they acknowledged, the presentation of the current draft should be significantly revised.

---

> > > ### Author Response · Authors · 2026-04-01
> > >
> > > We thank Reviewer 5M91 for engaging with our rebuttal and acknowledging that the additional information was beneficial. We fully agree that the presentation of the submitted draft needs revision, e.g. the broken appendix references and the placement of core equations in the appendix rather than the main text were clear shortcomings. These have been corrected in our working draft, though unfortunately ICML does not permit updated manuscripts during the rebuttal period.
> > >
> > > We would like to note for the record that the reviewer's substantive scientific concerns, regarding the expressiveness of the action space (Point 1), the motivation for the three capability domains (Point 2), and the need for quantitative evidence beyond LLM-as-judge correlations (Point 3), were each addressed in our rebuttal with detailed justifications and new controlled experiments, and were not contested in the reviewer's post-rebuttal response. The remaining concern pertains specifically to the presentation of the submitted draft.
> > >
> > > We believe these presentation issues are fully addressable in a camera-ready revision without any changes to the scientific contribution.

---

### Official Review · Reviewer_TW1h · 2026-03-13

**Soundness:** 3
**Presentation:** 3
**Significance:** 4
**Originality:** 3
**Overall Recommendation:** 5
**Confidence:** 4

**Summary:**

This paper introduces a simulated gig-economy environment for studying LLM-based agents in a labor market, operationalizing real-world dynamics such as a design for upskilling / exploiting tradeoffs, and bidding on jobs. The paper also provides simulations on specific market design problems that help to demonstrate its validity and usage, and a framework for interpreting agentic reasoning along the axes of metacognition, competitive awareness, and planning. The prompt scaffold explicitly defining these traits (SSA) outperforms alternative LLM prompting baselines (CoT, ReAct).

**Compliance With Llm Reviewing Policy:**

Affirmed.

**Key Questions For Authors:**

Questions above.

**Limitations:**

The limitations section is adequate. This may be covered by the acknowledgement of the stylized environment, but the scenario assumes deterministic upskilling, which is a particularly stylized portion. Aside: It could be interesting to run an ablation where upskilling is stochastic, or comes with additional costs outside of forgoing a round of pay.

**Strengths And Weaknesses:**

Strengths:
* The environment design is novel and appears rigorous - in particular, this paper provides a unique structure for evaluating delayed rewards and reputation-mediated consequences.
* The experimental ablations on market design (e.g. open vs. sealed bidding, flat-fee vs performance pay) are excellent and provide actionable insights for future market designers. It would be an interesting addition to hypothesize which additional potential simulated results in this setting may *not* transfer to traditional market findings. The grounding in established findings (e.g. wage deflation) help to validate the simulation mechanics.


Weaknesses:
* The metacognition measurement is defined as an agent's ability to measure its own latent, unobservable skill. However, the provided prompt in Appendix I explicitly provides the agent with its reputation score and job outcomes. How might we differentiate metacognition (e.g. inference such as 'My score is 4.0 but I've lost 3 successive jobs') vs. just reading its prompt?
* In the LLM-as-judge section, consider checking for structural biases between the SSA and LLM outputs.
  * https://arxiv.org/abs/2306.05685
  * https://aclanthology.org/2024.acl-long.511/


Nits: Fix appendix references (e.g. line 117 and 133), missing punctuation (e.g. period on 168), citation parentheses formatting throughout.

---

> ### Author Rebuttal · Authors · 2026-03-31
>
> We thank Reviewer TW1h for the thoughtful and supportive review, and for highlighting the novelty of the environment design, the delayed-reward / reputation-mediated setting, and the value of the market-design experiments.
>
> **1. Metacognition vs. merely reading prompt-visible statistics.**
>
> We agree this distinction needed to be clearer. In our rubric (Appendix L.2), simply restating prompt-visible information (e.g., "my coding reputation is 3.0*") receives at most a low score. Higher metacognition scores require cross-round inference about latent competitiveness from imperfect feedback. For example, from our traces:
>
> - _"My reputation is still 4.0*, but I have lost three consecutive mid-budget jobs at similar prices, so my effective competitiveness is lower than the public score suggests."_ — integrates reputation with a loss pattern to infer a gap not directly visible in any single round.
> - _"My coding reputation increased again, yet my earnings fell, which implies that pricing rather than skill is currently the bottleneck."_ — identifies a discrepancy between observable signals to diagnose the root cause.
>
> Here, the key criterion is not whether the agent can read a prompt field, but whether it synthesizes multiple signals across rounds to infer something about its latent state that no single prompt field states directly. We will clarify this distinction more explicitly in the rubric in future revisions.
>
> **2. Potential structural bias in the LLM-as-judge pipeline.**
>
> We thank the reviewer for the references, and will cite the suggested works + discuss possible structural bias in judge scoring as a limitation. To clarify, the trace-scoring analysis is applied to the baseline eight-model traces from Section 3.2, not to SSA-vs-baseline outputs, so the judge does not mediate the paper’s main SSA performance claim. Secondly, traces are anonymized and stripped of model/prompt identities before scoring. Lastly, there were some human validation with human annotators achieved inter-annotator agreement $\kappa=0.71$, and the judge–human correlation is $r=0.79$ on the composite score (per-dimension: $r=0.74$ metacognition, $r=0.81$ competitive awareness, $r=0.77$ planning). However, we agree with the reviewer that judge analysis should be treated as diagnostic rather than dispositive. Our primary evidence comes from the controlled prompting intervention and ablations, which compare actual market outcomes while holding backbone, observations, and action interface fixed.
>
> **3. Which findings may not transfer to real markets.**
>
> We appreciated this suggestion, and plan to discuss this further in revision. Some findings are likely directionally robust: for example, open pricing intensifying competition is consistent with online labor-market evidence [1], while performance-linked pay increasing training/effort aligns with standard incentive and human-capital theory. At the same time, AI-agent markets may differ qualitatively from human labor markets because agents are replicable, operate at much higher frequency, and may accumulate reputation much faster. These differences could amplify concentration and strategic specialization in ways not seen in traditional human markets. They may also change aggregate demand: if lower-cost agent labor makes many previously uneconomic microtasks worthwhile, total demand for labor-like services could increase rather than simply fall, in a Jevons-paradox-like manner. Our intent is therefore to position AI-Work as a mechanism-study environment for emerging AI-specific labor dynamics, rather than a literal extrapolation from human gig platforms.
>
> **4. Stochastic upskilling.**
>
> We agree this was important to probe, and thank the reviewer for the suggestion. We performed several new experiments for rebuttals, with a one-page summary [here](https://ibb.co/RpsZHk5D). In the robustness experiments, we included stochastic training ($p_{\text{train}}=0.8$) as part of a broader sweep over scoring, reputation, learning, and matching mechanisms. The main comparative-statics results still hold: open pricing lowers late-horizon winning bids, and performance-linked pay improves client utility. Interestingly, increased stochasticity dampens the training response. When training success is uncertain, agents face a noisier return on investment, which weakens the incentive to train relative to the deterministic case. We will discuss this in detail in future revisions, noting that training effects are the most mechanism-sensitive of our findings and should be interpreted with appropriate caution. Robustness result tables here: [Table 12](https://ibb.co/wrw0g1V3) and [Table 13](https://ibb.co/N6TT818F).
>
> All presentation nits (broken appendix references, citation formatting, punctuation, etc.) will be fixed in the revision.
>
> [1] Hong, Y., Wang, C. A., and Pavlou, P. A. “Comparing Open and Sealed Bid Auctions: Evidence from Online Labor Markets.” *Information Systems Research*, 27(1):49–69, 2016.

---

> > ### Author Rebuttal · Reviewer_TW1h · 2026-03-31
> >
> > Thanks for the feedback and acknowledgement.

---

### Official Review · Reviewer_GrSA · 2026-03-13

**Soundness:** 2
**Presentation:** 2
**Significance:** 3
**Originality:** 3
**Overall Recommendation:** 3
**Confidence:** 5

**Summary:**

This paper introduces AI-Work, a simulated gig economy platform where large language model (LLM) agents compete for jobs, build reputations, develop skills, and adjust strategies under uncertainty. The authors formalize the AI ​​labor market as a partially observable stochastic game with adverse selection and reputation dynamics. They benchmark 8 sota LLMs in this environment and identify 3 capability domains associated with agent success: metacognition, competitive awareness, and strategic planning. The paper proposes a cueing framework called “Strategic Self-Enhancing Agent (SSA)” to stimulate these reasoning patterns. SSA achieves a market share 2.3 times that of the standard cueing benchmark. The paper also investigates how platform design choices (open bidding vs. sealed bidding, fixed fees vs. performance-based pay) alter equilibrium behavior.

**Compliance With Llm Reviewing Policy:**

Affirmed.

**Final Justification:**

The author’s detailed rebuttal addressed some of my concerns.

**Key Questions For Authors:**

The main issues are listed in Weaknesses.  I am willing to increase my score if points 1 and 2 are resolved reasonably.

**Limitations:**

yes

**Strengths And Weaknesses:**

**Strengths:**
1. The AI ​​agent market is rapidly emerging, making it crucial to understand its microeconomic dynamics.

2. Environmental design aligns with economic principles, and market design experiments have yielded intuitive and economically sound results.

**Weaknesses:**
1. Many of the paper's conclusions heavily rely on stylized assumptions, such as the Cobb-Douglas client score, reputation update rule, matching, and training gain. Therefore, the resulting equilibrium is largely model-induced equilibrium, not necessarily robust equilibrium.

2. The agent groups in this paper are relatively small, with most experiments involving 8-16 agents. Given that the future AI labor market may involve much larger groups, and even many current economically practical multi-agent systems often exceed this scale, the simulation conclusions drawn in such a small market have limited relevance.

3. SSA is essentially a structured cue word, and the gains likely stem primarily from knowledge injection rather than capability activation. The conclusions in this paper do not control for the length and information content of the cue words.

---

> ### Author Rebuttal · Authors · 2026-03-31
>
> We thank Reviewer GrSA for the rigorous review and willingness to revisit the score if W1 and W2 are addressed. We conducted substantial new experiments and hope they sufficiently address all three weaknesses. A one-page summary of all new results: [image](https://ibb.co/RpsZHk5D)
>
> (For clarity: the missing “Appendix ??” references in the submission correspond to the formal environment/mechanism sections: scoring and matching in Appendix F.4/G, prompts in Appendix I.3, experiment settings in Appendix K.2, and full results in Appendix N - Fixed in revision.)
>
> **W1: Stylized assumptions / model-induced equilibrium.**
>
> We agree that the equilibrium could be driven by specific mechanism choices. Our goal in AI-Work is not to provide a high-fidelity model of labor markets, but to isolate a small set of economic forces in reputation-mediated gig markets under partial observability. Each mechanism is chosen as a tractable canonical baseline: Cobb-Douglas scoring captures the price-reputation tradeoff clients face, the discounted Beta-evidence reputation system provides recency weighting and principled cold-start handling, Gale-Shapley matching models platform-side optimization for client satisfaction, and saturating skill updates mirror standard diminishing-returns in human capital accumulation. We view our findings as **qualitative comparative statics of market design**, rather than predictions of real-world equilibria.
>
> To verify this, we varied the following parameters: scoring functions (CES with ρ in {-0.5, 0.5}, linear additive), reputation parameters (W=2, λ in {0.7, 0.85}, window H=10), learning (stochastic training, p=0.8), and matching (stochastic noise t=0.2), each crossed with open/sealed and flat/performance-pay interventions (5 seeds/configuration). Both core Section 3 findings hold across every perturbation tested: revealing prices lowers late-horizon winning bids in all conditions (0.711 to 0.608 aggregate, [results](https://ibb.co/wrw0g1V3)), and performance-linked pay improves client utility in all conditions (0.314 to 0.658, [results](https://ibb.co/N6TT818F)). The training response is directionally positive on average but more mechanism-dependent, varying with the persistence/noisiness of the reputation system. Because we perturb each mechanism class the reviewer highlighted (scoring, reputation, learning, matching), these effects are unlikely artifacts of a single parameterization or equilibrium construction.
>
> **W2: Scalability.**
>
> We evaluated six (N,K) configurations with jobs scaled proportionally to hold the job-to-agent ratio approximately constant (half SSA, half CoT, 5 seeds each). SSA maintains a reward advantage in all six settings (1.22x to 2.05x); at (128,4), SSA earns 317.7 vs. 181.1 for CoT (1.75x). While the advantage generally narrows at larger N due to intensifying competition, it persists in every configuration tested. Concentration dynamics from Section 3.1 also replicate with LLM agents at scale: at fixed K=4, increasing N from 8 to 64 raises Gini from 0.50 to 0.66; at fixed N=64, increasing K from 4 to 16 reduces Gini from 0.66 to 0.57. [Results table](https://ibb.co/v43fsJp4). This scale is comparable to current LLM-based economic simulations with fixed-policy agents (e.g., [1] uses N=100), while our setting uses adaptive agents under partial observability. These results suggest that strategic reasoning capabilities remain economically relevant even as market size grows, which is the central concern the reviewer raised.
>
> **W3: SSA as knowledge injection / prompt length bias.**
>
> We added two controlled experiments with the same backbone, observations, and action interface, varying only the scaffold. First, decomposing SSA into structure-only and knowledge-only variants ([prompts](https://ibb.co/MyQjNKq5)) shows structure contributes more than domain hints, with average market share by agent: 7.9% > 6.2% > 5.0% > 4.3% (full SSA > structure-only > knowledge-only > CoT, [results](https://ibb.co/9kzx625C)). Second, against approximately token-matched, alternatively structured prompts, the M/C/P decomposition outperforms all alternatives: SSA 6.1% vs. domain-relevant alternatives 4.2% avg ([prompts](https://ibb.co/J4Y4xvc)), CoT 3.6%, domain-irrelevant scaffolds 2.8% avg ([prompts](https://ibb.co/bR8snvC3), [results](https://ibb.co/kgfnBKfL)). Importantly, domain-irrelevant structured prompts perform at or below CoT, ruling out generic structure or longer prompts as the explanation. Together with the component ablation in Section 4.6 (each of M, C, P significant at p<1e-4), this suggests the gains are not explained by prompt length or generic knowledge injection, but by the alignment of the M/C/P scaffold with the market's information structure. Full write-ups will be included in the revised appendix.
>
> We hope these additions address the reviewer’s concerns and would be very grateful for any updated assessment.
>
> [1]Li et al. EconAgent, 2024, arxiv.org/abs/2310.10436

---

> > ### Author Rebuttal · Reviewer_GrSA · 2026-04-04
> >
> > Thank you for the detailed rebuttal. While it addressed some of my concerns, I have increased my score; however, my overall assessment remains negative. My main concern is that the market simulation is overly simplified and would require a more substantial, systematic revision of the paper to be adequately addressed.

---

### Official Review · Reviewer_t6s6 · 2026-03-17

**Soundness:** 3
**Presentation:** 2
**Significance:** 3
**Originality:** 3
**Overall Recommendation:** 4
**Confidence:** 3

**Summary:**

This paper constructs an AI market to study a microeconomic environment populated entirely by agents. Within this simulated market, agents compete for jobs, make strategic decisions, and adapt their behavior over time. Based on this setup, the paper analyzes the economic dynamics that emerge in agent-only labor markets and identifies the main factors associated with agent success.

**Compliance With Llm Reviewing Policy:**

Affirmed.

**Key Questions For Authors:**

No additional questions beyond the points above.

**Limitations:**

yes

**Strengths And Weaknesses:**

- Strengths:

  - The market modeling strikes a good tradeoff: it preserves key components while keeping the system from becoming overly complex, and it seriously balances validity and complexity.

  - The paper provides extensive analysis of the experimental results. It also offers a basic validation of the model and identifies several factors associated with agent success.

- Weaknesses:

  - Although the paper models the market carefully, real-world markets are still far more complex than this simulation, so the validity remains quite limited.

  - The presentation needs substantial revision.

    - There are many “Appendix ??” placeholders left in the paper.

    - The validation in Section 3.1 does not clearly tell readers where to find the detailed results in the appendix.

    - In Section 3.2, the model abbreviations are difficult to follow; it would be better to include the full model names in the main text.

    - The appendix should include a brief introduction to human gig-economy platforms to better support the realism of the simulated market.

---

> ### Author Rebuttal · Authors · 2026-03-31
>
> We thank Reviewer t6s6 for the feedback, and for recognizing the balance our model strikes between validity and tractability.
>
> We agree that real-world labor markets are much richer than our simulation. AI-Work is intended as a controlled microeconomic testbed to isolate a small number of core forces, e.g. reputation-mediated adverse selection, price competition, and skill investment under competition, in a setting where agent behavior and reasoning can be measured cleanly. We therefore interpret our results as identifying mechanisms and directional effects, rather than making quantitative claims about real-world equilibria.  In future revisions, we will make this scope more explicit.
>
> We also agree that the presentation should be improved, and will revise it directly in response to the reviewer’s comments:
> - All "Appendix ??" placeholders will be removed and properly linked.
> - Section 3.1 will more clearly direct readers to the validation construction and details.
> - Section 3.2 will use full model names/identifiers rather than relying only on shorthand abbreviations. (Detailed validation methodology/results are in Appendix K / N.1)
> - We will add a clearer appendix discussion connecting AI-Work to both human gig-economy platforms and early AI agent marketplaces.
>
> Concretely, this added discussion will summarize how AI-Work abstracts the same recurring economic loop seen on human gig platforms such as Upwork, Fiverr, Freelancer, and TaskRabbit, where clients post tasks with budgets, workers bid or offer fixed-price services, platform-side ranking depends on noisy public signals such as price/reputation/reviews, and completed work feeds back into future reputation and access to jobs. This also connects with a number of newly emerging AI-agent marketplaces (e.g. ClawGig, Dotblack, ClawTasks, WORQ), which implement many of the same economic mechanisms we simulate, in listings, bids, public profiles, reputation systems, repeated matching, and tradeoffs between short-run earnings and longer-run specialization. This helps clarify that AI-Work is a reduced-form model of a recurring market loop, rather than an imagined marketplace form, and also strengthens the paper’s realism motivation. More importantly, the emergence of these early agent-work sites suggests that AI labor markets are becoming practically relevant, which increases the value of studying their economic dynamics, strategic incentives, and failure modes in a controlled environment before deployment scales further.
>
> Beyond presentation, we also conducted additional post-submission experiments to strengthen the paper:
> - robustness experiments across alternative mechanism choices,
> - scalability experiments up to N=128 agents,
> - and prompt-control experiments testing whether SSA’s gains are explained by knowledge injection alone.
>
> A one-page summary of the new experiments is here: [image link](https://ibb.co/RpsZHk5D).
>
> We appreciate the reviewer’s feedback and will incorporate these clarifications and additions in the revised manuscript.

---

### Decision · Program_Chairs · 2026-04-30

**Decision:**

Accept (regular)

**Comment:**

Reviewers agreed that the paper addresses a timely issue—that is, the dynamics of the labor market in which agents compete for jobs. The paper presents a simulation test-bed for understanding such dynamics. Much of the discussion revolved around whether the proposed environment strikes the appropriate balance between validity and tractability of the abstractions. Given that reviewers found the framework solid and the analysis offered interesting and actionable insights, I believe the authors have struck the appropriate balance. Reviewers also raised concerns about the current presentation of the paper, but considering the authors’ commitment to addressing these issues, I don’t consider this a fatal problem. Overall, I believe this paper could be a valuable addition to the conference program.